# Strengthening government's response to COVID-19 in Indonesia: A modified Delphi study of medical and health academics

Yodi Mahendradhata[1,2]*, Trisasi Lestari[2], Riyanti Djalante[3]

1 Department of Health Policy and Management, Faculty of Medicine, Public Health and Nursing, Universitas Gadjah Mada, Sleman, Yogyakarta Province, Indonesia, 2 Center for Tropical Medicine, Faculty of Medicine, Public Health and Nursing, Universitas Gadjah Mada, Sleman, Yogyakarta Province, Indonesia, 3 Institute for the Advanced Study of Sustainability, United Nations University, Tokyo, Kanto, Japan

* ymahendradhata@ugm.ac.id

**Data Availability Statement:** All relevant data are within the paper and its Supporting Information files.

## Abstract

The Indonesian government has issued various policies to control COVID-19. However, COVID-19 new cases continued to increase, and there remain uncertainties as to the future trajectory. We aimed to investigate how medical and health academics view the Indonesian government's handling of COVID-19 and which areas of health systems need to be prioritized to improve the government's response to COVID-19. We conducted a modified Delphi study adapting the COVID-19 assessment scorecard (COVID-SCORE) as the measurement criteria. We invited medical and health academics from ten universities across Indonesia to take part in the two-round Delphi study. In the first round, participants were presented with 20 statements of COVID-SCORE and asked to rate their agreement with each statement using a five-point Likert scale. All participants who completed the first cycle were invited to participate in the second cycle. They had the opportunity to revise their answers based on the previous cycle's results and ranked a list of actions to improve government response. We achieved a moderate consensus level for five statements, a low consensus level for 13 statements and no consensus for two statements. The prioritization suggested that top priorities for improving the government's response to COVID-19 in Indonesia encompass: (1) strengthening capacity to ensure consistent, credible and targeted communication while adopting a more inclusive and empathic communication style to address public concerns; (2) ensuring universal access to reliable COVID-19 testing by expanding lab infrastructure, facilitating operational readiness, and scaling up implementation of proven alternative/complementary tests to RT-PCR; and (3) boosting contact tracing implementation capacity and facilitating contact tracing for all positive cases, involving key stakeholders in further development of the existing contact tracing system (i.e. PeduliLindungi) as well as its evaluation and quality assurance. Ultimately, our study highlights the importance of strengthening health system functions during the pandemic and improving health system resilience for dealing with future public health emergencies.

**Funding:** The authors received no specific funding for this work.

**Competing interests:** The authors have declared that no competing interests exist.

## Introduction

The COVID-19 pandemic is an unprecedented global crisis in the 75-year history of the United Nations [1]. As of September 24, 2021, there have been 230,418,451 confirmed cases of COVID-19, including 4,724,876 deaths, reported to WHO [2]. Most governments had not anticipated the rapid spread of COVID-19 and mainly were reactive in their crisis response [3]. Governments' failures to suppress the pandemic is disappointing and costly. One reason for failure to prevent the spread of SARS-CoV-2 is medical populism; politicians downplaying the pandemic, promoting simplistic solutions, or popularizing their responses to the crisis [4]. Many governments rely on vaccination against COVID-19, while priorities should also be given to well-tested public-health measures to control outbreaks [5]. Control measures must cover all health system building blocks, such as service delivery, medical products and technologies, health information systems, health workforce, financing leadership and governance [6]. COVID-19 response has highlighted the need for governments to improve their outbreak response by incorporating a comprehensive health-systems approach.

Indonesia is the world's largest archipelago and the world's fourth most populated country, with more than 260 million people; its size and diversity put it in a challenging position that very few other countries face [7]. The first two COVID-19 cases in Indonesia were confirmed on March 1, 2020, two months after SARS-CoV-2 was first reported on December 31, 2020, in Wuhan, China [8]. The initial claim of no case reported by Indonesia, before the first two confirmed cases, was questioned by many. During this period, the government did not issue travel restrictions and specific policies for quarantines of travellers coming in/coming back to Indonesia, despite reports of increasing COVID-19 cases in neighbouring countries [9]. The Indonesian government only issued a travel restriction on January 27, 2020, from Hubei province while evacuating 238 Indonesians from Wuhan.

After the initial and subsequent reports of infections as well as the first death reported from coronavirus on March 11, 2020, the Indonesian government realized the seriousness of the situation [8, 9]. Still, the large-scale social distancing regulation that restricts non-essential population mobility was only enacted a month after the first cases were reported and subsequently implemented in the capital, Jakarta, on April 10, 2020 [10]. The government has since issued various policies and actions to tackle COVID-19, such as appointing general hospitals as COVID-19 Referral Hospitals [9]. However, the trend of COVID-19 new cases continued to fluctuate. By September 24, 2021, Indonesia has reported 4,204,116 confirmed cases of COVID-19 with 141,258 deaths [2]. There remain uncertainties and concerns as to the future trajectory of COVID-19 in the country [7]. The impact on Indonesia's vulnerable health system, due to insufficient health workforce and health care infrastructure, will be devastating if COVID-19 continues the trajectory observed in other countries. Meanwhile, the economic, social, and non-COVID-19 related health system impact (e.g. disruptions of non-COVID-19 related health services) has already taken a significant toll on Indonesia.

Coordinated and comprehensive actions to suppress the COVID-19 pandemic in Indonesia need to be enhanced. The government needs to prioritize advice from medical and health professionals [4]. This study aims to consolidate advice from medical and health academics for the government to enhance the COVID-19 response in Indonesia. The study objectives are twofold. First, to investigate how medical and health academics view the government of Indonesia's handling of the COVID-19 epidemic. Second, to identify which areas of health systems need to be prioritized to improve the government's response to COVID-19 and strengthen health system resilience. This manuscript describes the modified Delphi method employed to achieve the study objectives, presents consolidated assessments by medical and health professionals on the government's COVID-19 response, ranked priorities for improvement, and elaborates their implications.

## Materials and methods

### Research design

We conducted a modified Delphi study to achieve the study objectives. The Delphi technique is a well-established approach to answering a research question by identifying a consensus view across subject experts [11]. This technique is commonly used in developing consensual guidance on best practice and exploration of a field beyond existing knowledge and the current conceptual framework [12]. The Delphi technique has been used internationally to investigate a wide variety of issues, including health issues. This technique is primarily used in health sciences by researchers when the available knowledge is incomplete or subject to uncertainty, and other methods that provide higher levels of evidence cannot be used [13]. In the context of COVID-19 in Indonesia, there are ongoing debates about whether the government can control the epidemic, whether proper measures have been taken, and whether the health system responds in a coordinated manner. Therefore, the Delphi method is suitable to build a consensus of experts on those issues. The modified Delphi method we used is similar to the standard Delphi method in terms of procedure (i.e., a series of rounds with selected experts) and intent (i.e., to arrive at consensus). The major modification consists of beginning the process with a set of carefully selected items.

We adapted the COVID-19 assessment scorecard (COVID-SCORE) as the measurement criteria for the modified Delphi study [5]. COVID-19 SCORE is a recently developed measurement tool that could be used to assess government accountability in managing the COVID-19 epidemic or other disease outbreaks. The tool was created based on the Pandemic Health System Framework elements adapted from the WHO's health system framework [6]. The COVID-SCORE consists of twenty policy statements about improving public health communication and health literacy, facilitating robust surveillance and reporting, developing pandemic preparedness, strengthening the health system, ensuring the health and social equity, and ensuring comprehensive confinement and de-confinement strategies. The twenty statements from COVID-SCORE were then modified into three types of questionnaires: Likert scale survey, short comment, and ranking scale. The investigators in this study reviewed the adopted COVID-SCORE statements, assessed content validity, construct validity and approved changes.

The adopted COVID-SCORE questionnaire was tested on six medical and health academics who were not working in the institutions participating in the study. Pilot test participants filled out the questionnaires and provided comments on face validity and clarity of statements at the pilot study's end. The questionnaire was then revised accordingly.

### Study population

We adopted the Cambridge Dictionary's definition of academic: a person who teaches in a college or university [14]. To be considered a participant in this study, each candidate had to fulfil the following criteria: (1) Hold a master or PhD degree in public health or clinical fields, (2) A researcher or lecturer in a reputable public university, and (3) Working at least two years at the selected institution.

We aimed to have regional representativeness of academics in the panel as Indonesia is diverse across its regions. Thus, we started with purposively sampling potential universities to represent four regions of Indonesia: Sumatera, Java, Kalimantan, and Eastern Indonesia. More than half of the Indonesian population lives on Java island, and most universities are also located in Java. Thus, a higher proportion of participants was allocated to universities in Java island. To ensure that respondents have the capacity and capability to participate in the survey,

Recruitment of universities

13 (2 Sumatera, 1 Kalimantan, 3 Eastern Indonesia, 7 Java)

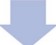

Participating universities

10 (1 Sumatera, 1 Kalimantan, 2 Eastern Indonesia, 6 Java)

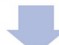

Invited academics

118 (12 Sumatera, 14 Kalimantan, 20 Eastern Indonesia and 72 Java)

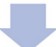

Participating academics in the first cycle

70 (7 Sumatera, 5 Kalimantan, 16 Eastern Indonesia, and 42 Java)

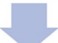

Participating academics in the second cycle

51 (4 Sumatera, 4 Kalimantan, 14 Eastern Indonesia, and 29 Java)

**Fig 1. The recruitment process for the modified Delphi study.**

we then performed snowball sampling by asking senior lecturers to recommend a list of academics from their respective universities.

The Delphi group size depends on group dynamics for arriving at consensus among experts rather than on statistical power [15]. However, a Delphi panel usually consists of 15 to 30 participants from the same discipline, or five to 10 per category from different professional groupings [16]. Since the Delphi method is an iterative online process, there is a high risk of non-respond participants at every study phase. To ensure enough participants were available at the last stage, we recruited at least double the number of participants needed at the last stage (about 30). An overview of the recruitment process is presented in **Fig 1**.

## Data collection

We sent an invitation by email to 118 academics from 10 universities in Indonesia to participate in the first round of the online Delphi Survey. The email contained a short description of the background and aims of the study, the process of two-cycle Delphi surveys, and a link to an online questionnaire on the Qualtrics Research Suite platform. The survey questionnaire is available from the authors on request.

In the first round, participants were presented with 20 statements of COVID-SCORE and asked to rate their agreement with each statement using a five-point Likert scale: (1)

completely disagree, (2) disagree, (3) neutral, (4) agree, and (5) completely agree. All statements were written in Bahasa Indonesia and English to minimize variations in interpretations among participants. Participants could optionally add short comments or reflections relating to each statement in free text format. Participants had seven days (20 to July 26 2020) to fill out the questionnaire. A reminder was sent two days before the due date, and we extended the first cycle for one day. A total of 71 academics participated in the first cycle.

All participants who completed the first cycle were invited to participate in the second cycle of the Delphi survey, which took place between July 28 and August 3 2020. Summary of results from the first cycle was re-administered to all participants. Analyzed data reflected a more objective view on each statement and provided more confident and solid direction for panel review. After reading the results, participants had a chance to revise their initial responses, so the end results should be closer to a satisfactory degree of consensus. All experts in the panel were kept anonymous to each other but not to the researcher throughout the process. Based on the result of the first cycle, participants were asked to rank the priority of actions to improve government response to the COVID-19 pandemic. A reminder was sent two days before the due date, and a final reminder was sent on the last day of data collection. The result of the second (final) cycle was analyzed, and feedback to all participants a week later.

## Analysis

The dataset and code book has been made available as supplementary files to this manuscript. Participant profiles, i.e. age, gender, profession, and region, were coded. We used percent agreement, which is the most common definition for consensus in Delphi [17]. We calculated percentages of score for each statement. "High consensus level" was defined as ≥95% agreement in the rating of the single statements by the panellists; "Moderate consensus level" was defined as the agreement of 75–94%, "Low consensus level" was defined as the agreement of 50–74%, and "No consensus" was recorded if the agreement was <50%. An agreement was if the panellists replied either "agree" or "strongly agree" and disagreement if "disagree" or "strongly disagree". In the second cycle, respondents had the opportunity to revise their answers. We calculated p-values to assess the significance of the change in the degree of consensus for each statement between the cycles. Although it was not compulsory, the panellists also had the opportunity to comment on the statements or reasons for their assessment. For the ranking score, each priority action was be scored according to its sum of rank. The lowest total score was considered the highest priority. After completing the preliminary analysis, all experts in the panel were invited to a webinar to discuss the research brief, which summarized the process and interpretation of findings from the Delphi survey. The webinar was delivered via Zoom, facilitated by the authors and completed in 90 minutes. During the webinar, respondents were asked whether they agreed with the presented results and priority ranking. The authors documented their responses, and the entire webinar was audio and video recorded.

## Research ethic

Written informed consent was required before a participant could start filling the online questionnaire. Anonymity is one of the main features that characterized this study method. Participants did not meet and did not get any information about other participants involved in the study. Therefore, they could freely submit their ideas, unbiased by the identities or pressures of others. All participants were coded in the analysis; thus, their opinions and comments were anonymous to the investigators. The study was reviewed and granted ethical approval by the Medical and Health Research Ethics Committee Faculty of Medicine, Public Health and Nursing Universitas Gadjah Mada/Dr. Sardjito General Hospital (Protocol ID KE/FK/0743/EC/2020).

## Results

Out of 118 academics invited to participate in this study, 75 responded to the invitation. Two individuals declined to participate, and three did not finish the survey. In total, 70 academics from 10 public universities completed the first cycle of online Delphi survey (response rate 59%), ensuring representation from four Indonesia's regions (Sumatera: 1 university, 7 participants (10.0%); Kalimantan: 1 university, 5 participants (7.1%); Eastern Indonesia: 2 Universities, 16 participants (22.8%); and Java: 6 universities, 42 participants (60.0%)). Gender was equally represented in this study (35 males and 35 females). Age ranged from 27 to 71 years and median 46 years. Most of the participants hold a doctoral degree (48, 68.5%) and the rest hold a master degree (22, 31.5%). Most participants were from the public health field (48, 69.0%), and the rest were from the clinical field (12, 16.9%), health nutrition (5, 7.0%), nursing (3, 4.2%) and biomedical science (3, 4.2%). Most participants (53, 75.7%) were not directly involved with the COVID-19 response team. Respondents didn't need to comment on each Likert statement, but 50 (71%) were willing to share their opinions.

We sent an invitation to participate in the second cycle of the Delphi survey to 68 respondents. Two respondents from the first cycle were excluded because they did not provide contact details. It was recorded that 58 respondents accessed the survey, but only 51 completed the survey (75.0% response rate). About 60% of respondents (31 individuals) revised their response to Likert scale questions, and 47 (92%) proposed priority rank of actions. The result of the online Delphi survey is summarized in **Table 1**. Out of 20 survey statements, we achieved a moderate consensus level for five statements, 13 low consensus levels, and no consensus for two statements. No statements received a high consensus level. Compared to the first cycle, the consensus degree was improved for 15 statements and reduced for four statements. However, a significant change ($p < 0.05$) of consensus degree between the cycles was only observed for one statement ("Mental health outreach services have been expanded to meet increased demand"). The majority of respondents agreed to 11 statements and disagreed with nine statements. During the webinar to discuss these results, experts confirmed their agreement with the presented results and priority ranking. They also suggested recommendations for follow-ups of the study. Accordingly, the results were shared with the National Planning Development Agency and informed the development of a follow-up study. Some of the results of this subsequent study have now been published elsewhere [18]."

### Statements with moderate consensus level

Moderate consensus level was achieved for the following statements: (1) the government has maintained a partnership with the WHO, other countries and international NGOs in responding to the pandemic (87.5% agree), (2)the government has tried to address the health and socioeconomic impact of instituting and easing containment measures (75.5% agree), (3) contact tracing is implemented for positive cases (75.7% agree); (4) everyone can get a free, reliable COVID-19 test quickly and receive the results promptly (78.6% disagree), (5) Mental health outreach services have been expanded to meet increased demand (77.1% disagree).

**Government collaboration with other countries, WHO and international bodies.**   Most respondents agreed that the government maintained a partnership with the WHO, other countries, and international NGOs to respond to the pandemic (moderate consensus level, 87.5% agree). However, several respondents criticized that the form of partnership is unclear and lacks publication about activities done for the partnership. The range of collaboration was also not broad enough, and a respondent suggested expanding the cooperation with countries in Europe or the United States. It was also recommended that adaptation of international

Table 1. Consensus on policy statements and ranked priorities.

| No | COVID-SCORE statements | First cycle (N: 70) | | | | Second cycle (N: 70) | | | | Interpretation | p-value | Rank | |
|---|---|---|---|---|---|---|---|---|---|---|---|---|---|
| | | Agreement | Neither agree nor disagree | Disagreement | Agreement / disagreement proportion | Agreement | Neither agree nor disagree | Disagreement | Agreement / disagreement proportion | | Significance change | Total score | Rank |
| 1 | The authorities communicate clearly and consistently about COVID-19 and provide public health grounds for their decisions. | 26 | 10 | 34 | 48.5% disagree | 24 | 10 | 36 | 51.4% disagree | Low consensus level | 0.37 | 224 | 1 |
| 2 | Government communications target the entire population in all its diversity (e.g. language, culture, education, and socioeconomic level). | 20 | 10 | 40 | 57.1% disagree | 14 | 8 | 48 | 68.5% disagree | Low consensus level | 0.08 | 305 | 5 |
| 3 | Public health experts, government officials, and academic researchers agree on COVID-19 nomenclature and clearly explain the reasons for public health measures | 23 | 18 | 29 | 41.4% disagree | 19 | 13 | 38 | 54.2% disagree | Low consensus level | 0.07 | 288 | 4 |
| | | First cycle (N: 70) | | | | Second cycle (N: 70) | | | | Interpretation | p-value | Rank | |
| | | Agreement | Neither agree nor disagree | Disagreement | Agreement / disagreement proportion | Agreement | Neither agree nor disagree | Disagreement | Agreement / disagreement proportion | | Significance change | Total score | Rank |
| 4 | Everyone can get a free, reliable COVID-19 test quickly and receive the results promptly | 6 | 10 | 54 | 77.1% disagree | 7 | 8 | 55 | 78.6% disagree | Moderate consensus level | 0.41 | 267 | 2 |
| 5 | Contact tracing is implemented for positive cases | 46 | 11 | 13 | 65.7% agree | 53 | 9 | 8 | 75.7% agree | Moderate consensus level | 0.10 | 279 | 3 |
| 6 | Public health bodies maintain robust national, subnational, and local epidemiological databases, updated and reported daily | 47 | 10 | 13 | 67.1% agree | 50 | 11 | 9 | 71.4% agree | Low consensus level | 0.29 | 379 | 8 |

(Continued)

Table 1. (Continued)

| No | COVID-SCORE statements | First cycle (N: 70) | | | | Second cycle (N: 70) | | | | Interpretation | p-value Significance change | Rank Total score | Rank |
|----|----|----|----|----|----|----|----|----|----|----|----|----|----|
| | | Agreement | Neither agree nor disagree | Disagreement | Agreement / disagreement proportion | Agreement | Neither agree nor disagree | Disagreement | Agreement / disagreement proportion | | | | |
| 7 | There are enough qualified health workers and medical equipment (e.g. ventilators and face masks) to meet national needs | 21 | 10 | 39 | 55.7% disagree | 18 | 5 | 47 | 67.1% disagree | Low consensus level | 0.08 | 354 | 7 |
| 8 | The government can require private manufacturers to produce critical equipment rapidly, if needed | 50 | 10 | 10 | 71.4% agree | 50 | 9 | 11 | 71.4% agree | Low consensus level | 0.5 | 518 | 11 |
| 9 | A pandemic preparedness team that includes public health and medical experts is coordinating the national response | 40 | 16 | 14 | 57.1% agree | 39 | 14 | 17 | 55.7% agree | Low consensus level | 0.43 | 382 | 10 |
| 10 | Infection prevention and care guidelines and protocols are comprehensive and up to date | 41 | 21 | 8 | 58.6% agree | 43 | 20 | 7 | 61.4% agree | Low consensus level | 0.2 | 347 | 6 |
| 11 | Health systems have sufficient funding and infrastructure to care for all COVID-19 patients. | 13 | 17 | 40 | 57.1% disagree | 12 | 13 | 45 | 64.3% disagree | Low consensus level | 0.19 | 380 | 9 |
| 12 | Everyone has uninterrupted access to regular health services | 16 | 16 | 38 | 54.3% disagree | 14 | 11 | 45 | 64.3% disagree | Low consensus level | 0.11 | 554 | 13 |
| 13 | Primary care services and social services are coordinating and collaborating with each other during the pandemic | 25 | 12 | 23 | 35.7% agree | 23 | 23 | 24 | 34.3% disagree | No consensus | 0.43 | 562 | 14 |
| 14 | Mental health outreach services have been expanded to meet increased demand | 11 | 26 | 33 | 47.1% disagree | 11 | 12 | 37 | 77,1% disagree | Moderate consensus level | 0.00 | 760 | 20 |

(Continued)

**Table 1.** (Continued)

| No | COVID-SCORE statements | First cycle (N: 70) | | | | Second cycle (N: 70) | | | | Interpretation | p-value | Rank | |
| --- | --- | --- | --- | --- | --- | --- | --- | --- | --- | --- | --- | --- | --- |
| | | Agreement | Neither agree nor disagree | Disagreement | Agreement / disagreement proportion | Agreement | Neither agree nor disagree | Disagreement | Agreement / disagreement proportion | | Significance change | Total score | Rank |
| 15 | Task-sharing, task-shifting and telehealth are being used to optimize the delivery of health care services | 26 | 20 | 24 | 52% agree | 29 | 17 | 24 | 52.9% agree | Low consensus level | 0.46 | 651 | 18 |
| 16 | Appropriate measures have been taken to protect members of vulnerable groups such as the elderly, the poor, migrants, and the homeless | 16 | 16 | 38 | 54.3% disagree | 10 | 27 | 43 | 61.4% disagree | Low consensus level | 0.19 | 543 | 12 |
| 17 | COVID-19 efforts are focused on densely populated, low-resource areas | 23 | 20 | 27 | 38.6% disagree | 19 | 20 | 31 | 44.3% disagree | No consensus | 0.25 | 625 | 16 |
| 18 | Public health measures have been implemented to protect people in institutions and other confined settings | 34 | 15 | 21 | 61.8% agree | 33 | 10 | 27 | 55,0% agree | Low consensus level | 0.22 | 676 | 19 |
| 19 | The government is addressing the health and socioeconomic impacts of instituting and easing containment measures | 34 | 25 | 11 | 77.3% agree | 37 | 11 | 12 | 75,5% agree | Moderate consensus level | 0.41 | 611 | 15 |
| 20 | The government is collaborating with other countries, WHO, and other international bodies in responding to the pandemic | 48 | 13 | 9 | 84.2% agree | 49 | 14 | 7 | 87,5% agree | Moderate consensus level | 0.30 | 638 | 17 |

guidelines or recommendations must be adjusted to local context and availability of resources in the country.

**Contact tracing for positive cases.** Most participants agreed that contact tracing had been implemented, and it ranked third on priority action recommendation (moderate consensus level, 75.7% agree). We received 37 comments on contact tracing. Out of it, 28 respondents expressed their concerns regarding delayed initiation, coverage of contacts screened, unclear procedure, implementation variation at the local level, coordination and resources needed when caseload continues to increase. There was also the issue of patient honesty and induced stigma due to staff visits to the contact's house. Six respondents mentioned that contact identification and tracing was not yet done for some positive cases.

**Addressing the health and socioeconomic impacts of containment measures.** Half of the respondents agreed that "the government had tried to address the health andsocioeconomic impact of instituting and easing containment measures (moderate consensus level, 75.5% agree). The respondents were aware that the government provided incentives and financial relief for vulnerable people affected by the pandemic. However, several respondents criticized that the measures were more focused on social and economic impact than health. Several respondents also suggested that weak monitoring and easing of confinement measures have created a false perception that the situation is improving. there were no sanctionsfor people who did not wear a face mask or did not adhere to infection prevention measures when in public.

**Free, reliable COVID-19 test.** Almost all respondents disagreed that the COVID-19 test was available for free, and the result could be received promptly (moderate consensus level, 78.6% disagree). Respondents acknowledge that the COVID-19 test is available for free for those with COVID-19 symptoms or who have immediate contact with a COVID-19 patient. However, people who didn't meet the diagnostic criteria must pay for any COVID-19 tests. At the pandemic's beginning, diagnostic kits were scarce, and many people who wanted to be tested were rejected. Local government often conducted free mass screening but only in selected areas, such as markets and confined settings. When people got tested, they often have to wait for days or weeks before getting the test result. There were also concerns about the sensitivity and specificity of rapid tests available in many healthcare facilities. A rapid test is often needed for administrative purposes, e.g. when people want to travel to another province, they need to show a rapid test result.

**Mental health outreach services.** Only 15% of respondents agreed that mental health outreach service was expanded during the pandemic (moderate consensus level, 77.1% disagree). The majority of respondents commented that they were not aware such a service existed. However, many respondents agreed that mental health outreach service is needed and should be expanded to reach people at high risk of suffering from mental illness due to the crisis. Information about the availability of mental health services should be shared with the community.

## Statements with a low consensus level

Low consensus level was achieved for the following statements: the government has tried to maintain robust epidemiological databases at national and local levels (71.4% agree); the government should involve in country private manufacturers to produce critical equipment rapidly as part of corporate social responsibility (67.1% agree); a pandemic preparedness team that includes public health and medical experts is coordinating the national response (55.7% agree); infection prevention and care guidelines and protocols are comprehensive and up to date (61.4% agree); task sharing, task shifting and telehealth are being used to optimize the delivery of health care services (52.9% agree); public health measures have been implemented

to protect people in institutions and other confined settings (55.0% agree); the government has target the entire population (68.5% disagree), there were enough qualified healthcare workers and medical equipment to meet national needs (67.1% disagree); the government has communicated clearly and consistently about COVID-19 (51.4% disagree); the government has enough funding and infrastructure to care for all COVID-19 patients, particularly in the long run (64.3% disagree); public health experts, government officials, and academic researchers agree on COVID-19 nomenclature and have clearly explain the reason for public health measures (54.2% disagree); access to regular health services was uninterrupted (64.3% disagree); and that appropriate measures have been taken to protect members of vulnerable groups, such as the elderly, the poor, migrants, and the homeless (61.4% disagree).

**Updated and reported epidemiological databases.** It was acknowledged that the government had tried to maintain robust epidemiological databases at national and local levels (low consensus level, 71.4% agree). It ranked eighth on priority recommended actions. There were 36 comments on this issue which mainly indicate that epidemiological data was available and accessible. Nevertheless, respondents also highlighted the importance of accuracy, validity, accessibility, transparency, consistency, timely reporting, synchronization, utilization, presentation and interpretation of data. It is believed that these issues have created confusion in the community and affected public trust in published epidemiological data.

**Private manufacturers roles in the pandemic.** Respondents generally agreed that the government can require private manufacturers to produce critical equipment rapidly. The government has involved domestic private manufacturers to produce critical equipment rapidly (low consensus level, 67.1% agree). It ranked 11 on priority recommendation action. Respondents who provided comments also indicated that the central and local governments had not optimized the opportunity to collaborate with private manufacturers. The initiative to involve the private sector was considered to start too late. Concern on product quality was also raised. Thus, guidance and supporting policy are needed to attract investors to collaborate with the government and ensure public trust in quality of products. A respondent suggests focusing on manufacturing and expanding access to a fast, efficient, accurate and affordable diagnostic tool for COVID-19.

**Pandemic preparedness team.** Most respondents agree that the pandemic preparedness team has included public health and medical experts (low consensus level, 55.7% agree). It ranked tenth on the list of recommended actions. Implementation, however, is varied between regions, depending on local governments. Three respondents mentioned that the involvement of public health experts only started a month before the survey, which was considered late. Respondents also identified public health experts who provided recommendations personally and did not represent their institution. Recommendations from public health experts were not always accepted and resulted in inappropriate policy from the public health point of view. Thus, the extent to which public health experts have been involved in the response team remains questionable. Several respondents highlighted the importance of improving coordination and collaboration of the response team with public health and medical experts at the national and local levels.

**Infection prevention and care guideline.** The majority agreed that infection prevention and care (IPC) guidelines are available and updated (low consensus level, 61.4% agree). The Ministry of Health had released the 5th revision of the IPC guideline [19] when the survey was conducted. Comments on the IPC guideline were mainly related to the frequent update of the guideline but not followed by appropriate dissemination and training of the new guideline. Various institutions and professional organizations were also releasing IPC guidelines, potentially inconsistent with the national IPC guideline. Monitoring and supervision for the IPC

implementation also had not existed, and in many areas, materials needed for infection control was scarce. All of these led to the diverse application of IPC guidelines in the field.

**Task sharing, task shifting, and telehealth.** Less than half of respondents agreed that task-sharing, task-shifting and telehealth were used in healthcare services (low consensus level, 52.9% agree). In many health facilities, particularly private health facilities, telehealth has been used for some time. However, respondents were concerned that a policy to guide the implementation of a wide range of telehealth services was unavailable. Reporting and recording a patient case using the telehealth system was not standardized, and resources to run telehealth were not available in many areas, particularly in the rural area. Payment for telehealth services was also another issue that needed to be discussed by policymakers. While telehealth is generally understood, several respondents commented that task-sharing and task-shifting were not well implemented and rarely discussed.

**Protection of people in institutions and other confined settings.** Almost half of the respondents agreed that public health measures had been implemented to protect people in institutions or other confined settings (low consensus level, 55.0% agree). Local governments have published policies to guide people who work from offices or other working places. Intervention includes a mandatory face mask, social distancing, and hand washing. Several respondents mentioned that the protection of people in institutions depends on institutions policy, and implementation was varied. There was no evaluation of whether the policy was implemented, and there was no sanction for breaking the rules. One respondent highlights the importance of adjusting the ventilation system to ensure safe air circulation in a building.

**Communication of COVID-19.** Half of the respondents disagreed that the government has communicated clearly and consistently about COVID-19 (low consensus level, 51.4% disagree). Respondents acknowledged that the government had used various communication channels to provide information on COVID-19 to the community [20]. Several official websites provide updated daily caseload data, policy and recommendations and have become the primary source of information. However, many respondents criticized inconsistencies of information, particularly in the early pandemic phase. Some ministries and institutions provided conflicting information and created confusion in the community. Information was often unclear, ambiguous and the public health background of policy decisions was often not explained. All of these lead to various interpretations and confusion, even among highly educated people. The presentation of data was monotone and didn't show a sense of urgency. Lack of coordination in the delivery of information was also felt. A respondent suggested that the government create innovation in delivering information to ensure its rapid distribution to the whole population. Since establishing a national response team, communication has been getting more coordinated, precise, and consistent.

**Communications target the entire population in all its diversity.** Most respondents did not agree that the government has targeted the entire population (low consensus level, 68.5% disagree). Besides Indonesia being the largest archipelagic country globally with hundreds of tribes and languages, respondents also raised the importance of delivering information to the poor and disabled. Various institutions have produced health campaigns, and several campaigns have been made using the local language. Respondents suggested that campaigns should adapt local wisdom and use language that should be easy for people with low literacy to understand. The use of the internet and social media to share information has discriminated population living in rural areas without access to the internet or television. Some respondents also doubted that the flow of information had reached the lowest level of community group (the neighbourhood or RT/RW). There was a concern that information only reached those with a direct link to the response team. Thus, it is essential to involve community or religious leaders in delivering information.

**COVID-19 nomenclature.** More than half of respondents disagreed that public health experts, government officials, and academic researchers agreed on COVID-19 nomenclature and has clearly explained the reason for public health measures (low consensus level, 54.2% disagree). Respondents commented that many people did not understand the nomenclature regarding the different terms, people without symptoms (OTG), people in surveillance (ODP) and patient on surveillance (PDP), and what level of surveillance entails for each category. This issue was not only faced by the community but also by officials who worked directly for COVID-19. Another confusion started when the nomenclatures and definitions were updated into eight new terms in mid-July. There were also issues regarding terms used for infection control, disease transmission and treatment. The government needs to create a strategy for effectively informing the community about updated terminology and its use.

**Availability of qualified health workers and medical equipment.** The majority of respondents disagreed that there were enough qualified healthcare workers and medical equipment to meet national needs (low consensus level, 67.1% disagree). Medical personal protective equipment was scarce in every part of the country, and prices were skyrocketing after months of COVID-19. With an increasing number of cases, hospital beds and ventilators will not be enough. The shortage of PPE and medical equipment is more pronounced outside Java, where more hospitals are located in rural areas. An increasing number of infected health workers is concerning, especially when the caseload is also increasing. In previous outbreaks, there were special hospitals assigned to manage cases. Thus, many hospitals were not prepared to handle infectious disease pandemics. Many healthcare workers were not appropriately trained and competent to handle infectious disease patients.

**Funding and infrastructure for COVID-19.** Majority of respondent disagreed that the government have enough funding and infrastructure to care for all COVID-19 patients, particularly in the long run (low consensus level, 64.3% disagree). Several respondents commented that funding is available and that the government has tried to fulfil the need for health infrastructure, but the use of funds was hampered by bureaucracy. Delays in the payment of hazard incentives for frontline healthcare workers hints that the procedure to release the funds is still complicated.

**Access to regular health services.** Most respondents disagreed that access to regular health services was uninterrupted (low consensus level, 64.3% disagree). During the pandemic, many healthcare facilities limited their services by reducing opening hours, reducing daily patient quota, postponing elective surgery, and lengthening the interval of a follow-up visit for chronic disease patients. On the other hand, many people were afraid to visit healthcare facilities for fear of exposure to COVID-19 cases and the stigma associated with COVID-19. Several respondents commented that access to healthcare was available as usual, but the community was not well informed about it.

**Protection of vulnerable groups.** Only ten respondents agreed that appropriate measures had been taken to protect vulnerable groups, such as the elderly, the poor, migrants, and the homeless (low consensus level, 61.4% disagree). The government have released financial incentives for the poor and encouraged older adults and young children to stay at home. Many non-governmental organizations and philanthropies raise funds to support those affected by the COVID-19 pandemic, including the poor, elderly, and frontline healthcare workers. But many respondents commented that specific intervention or campaign targeting vulnerable population is still inadequate and can be optimized.

## Statements with no consensus

There was no consensus on whether primary care services and social services are coordinating and collaborating during the pandemic and whether COVID-19 efforts are focused on densely populated, low-resource areas.

**COVID-19 efforts on densely populated, low resource area.** Almost half of the respondents disagreed that COVID-19 efforts are focused on densely populated, low resource areas (no consensus, 44.3% disagree, 27.1% agree). Many respondents were not aware that the government has been focusing its efforts on this specific population. However, many agreed to this recommendation due to the high risk of transmission in such a setting. Implementation of this recommendation will depend on the local government decision.

**Coordination and collaboration of primary care services and social services.** Only one-third of respondents agreed on coordination and collaboration between primary care and social services (no consensus, 34.3 disagree, 32.8% agree). Many NGOs are actively involved in COVID-19, and primary healthcare facilities usually collaborate with social services in their coverage area. However, several respondents commented that the role of primary healthcare and social services could be optimized through better coordination and collaboration. The local government plays an essential role in building this collaboration, as shown in several provinces.

## Discussion

Our study highlighted moderate consensus among Indonesian medical and health academics regarding that: (1) the government has maintained a partnership with the WHO, other countries and international NGOs in responding to the pandemic; (2) the government has tried to address the health and socioeconomic impact of instituting and easing containment measures; (3) contact tracing has been implemented for positive cases; (4) Not everyone can get a free, reliable COVID-19 test quickly and receive the results promptly; and (5) Mental health outreach services have not been expanded to meet increased demand.

Our findings accordingly highlighted that the top priorities for improving ' 'government's response to COVID-19 in Indonesia encompass: (1) The authorities communicate clearly and consistently about COVID-19 and provide public health grounds for their decisions; (2) Everyone can get a free, reliable COVID-19 test quickly and receive the results promptly; (3) Contact tracing is implemented for positive cases; (4) Public health experts, government officials, and academic researchers agree on COVID-19 nomenclature and clearly explain the reasons for public health measures; and (5) Government communications target the entire population in all its diversity (e.g. language, culture, education, and socioeconomic level). These recommendations are discussed in more detail in the following paragraphs.

Communication was considered critical for effective government response by our Delphi study participants, highlighting three communications goals within the top five priorities (Priority rank 1; 4 and; 5). There have been numerous government communication problems across multiple countries, including Indonesia [21]. For instance, ineffective risk communication impeded the emergency response in Wuhan's outbreak management [22]. The Wuhan government did not integrate scientific risk communication into policy decisions, and the local government even delayed reporting. It handled the information publicity in an ambiguous way which diminished public perception associated with COVID-19.

The only source of official information at the early phase of the pandemic in Indonesia came from the presidential office and the Indonesian Health Ministry. The government did not appear to have a "sense of crisis" even as early as February 2020. This was exacerbated by "antiscience" statements issued by government officials [23]. The government appointed a spokesman who informed the spread of COVID-19 through one door regularly twice a day. However, when the number of cases increased in several areas, mayors and governors also issued statements, which often contradicted the official data held by the presidential office and Minister of Health [24].

Communication discordance signifies the failure of governmental systems, which significantly diminishes public trust in the government and considerably increases public confusion and fear about COVID-19 risks [21]. It has often been challenging for the public to differentiate between evidence-based and less scientifically reliable information during the COVID-19 pandemic, partly due to poor communication by government officials [25].

Arguably, Indonesia and many other countries have paid high costs due to COVID-19, which could have been prevented and addressed more effectively if governments had more responsive and strategic risk communication [21]. Communication is a substantial action, not just a precursor to action [26]. Policymakers should cautiously consider the quality of information circulated through private sources and social networks [27]. The Indonesian government thus needs to improve their efforts to disseminate information on the pandemic and employ strategies for improved communication management to citizens through social media and mainstream information sources [28].

The Indonesian government has implemented government-centred public communication without prohibiting media, including social media, from spreading general information about COVID-19 [29]. Thus, the primary sources of information for the public during the pandemic are social media, online media, and mass media. The most popular sources for preventive measures are social media (83.6%), television (78.5%), and WhatsApp (76.0%) [30]. Notably, Indonesia faces problems with the spread of misleading and harmful messages that hamper COVID-19 prevention efforts. The Ministry of Communications and Informatics monitors the spread of COVID-19 related mis/dis-information daily, with records of 1,550 hoaxes related to Covid-19 and 177 hoaxes related to Covid-19 vaccines from January 2020 to April 2021 [31]. In response, the government has partnered with digital technology-based industries and social media influencers for disseminating information [32]. Notwithstanding, merely sharing situation updates and policies may be inadequate to capture public attention [33]. Consistent, credible and targeted communication is critical in encouraging people to comply with COVID-19 control measures [25].

Our Delphi participants also emphasized the importance of enhancing testing and contact tracing. A test, trace and isolate strategy remains the most effective method of controlling the COVID-19 outbreak until an effective vaccine has been developed [34]. The participants suggested that the government prioritize efforts to ensure that everyone can get a free, reliable COVID-19 test promptly and rapidly receive the results. Timely and accurate laboratory testing is essential to manage the COVID-19 pandemic [35]. Serological tests have the main shortcoming of a late positivity during the disease course, although attractive due to their lower cost and ease of implementation [35]. Reverse transcription-polymerase chain reaction (RT-PCR) thus remains the gold standard for SARS-CoV-2 diagnosis, but several operational issues significantly limit the test use. Indonesia has been facing a hard time to rapidly scale-up laboratory capacity for RT-PCR [36]. Although the number of reference labs increases, the daily number of testing results remains fluctuated and unstable.

The gap between the RT-PCR testing capacity in laboratories and the number of suspected cases to be tested remains significant even one year after the first case was reported. Accordingly, the government adopted the WHO recommendation to use antigen detection rapid diagnostic test (Ag-RDT) to diagnose SARS-CoV-2 infection in settings where RT-PCR is unavailable or where prolonged turnaround precludes clinical utility [37]. The utilization of Ag-RDT has augmented the country's testing capacity. As of September 19 2021, all provinces in Indonesia have managed to meet the WHO recommended testing rate of at least one person per 1000 population per week [38]. Additionally, to enhance capacity to detect variants of concerns, the Ministry of Health has incorporated 21 institutions into its SARS-CoV-2 genome surveillance laboratory network, with the capacity of around 1,750–2,700 tests per month. As

of August 29 2021, Indonesia has shared 5,755 (0.142% cases) sequences since the beginning of the pandemic, according to GISAID (Global Initiative on sharing all Influenza data). By August 2021, the Ministry of Health announced that the Delta Variant had been detected in six provinces, i.e. DKI Jakarta, West Java, East Kalimantan, Central Java, East Nusa Tenggara, and Bali 1 2021 [39]. According to medical and health academics, in complement to enhancing testing, contact tracing for all positive cases should be a top priority for the Indonesian government. Contact tracing prevents transmission of infectious diseases by identifying, assessing and managing people who have been in close contact with an infected individual [33]. A recent mathematical modelling exercise highlighted that contact tracing is a critical component of the most rational scenario in controlling COVID-19 in Indonesia [40]. Another modelling exercise estimated that a high proportion of contacts must be successfully traced to ensure an effective reproduction number lower than 1 in the absence of other measures [41]. Combined with moderate physical distancing measures, contact tracing and self-isolation would be more likely to achieve control of SARS-CoV-2 transmission. Contact tracing has been used extensively in previous emerging infectious disease outbreaks [42]. Recent studies suggested that the effectiveness of contact tracing could also be enhanced through app-based digital tracing [43].

The Indonesian Ministry of Information and Communication launched a mobile application called PeduliLindungi. The app enables users to compile data related to the spread of COVID-19 in their communities to help bolster the Indonesian government's efforts to trace and track confirmed cases. Users are expected to register as participants, share their travelling locations, and trace whether they have contacted persons exposed to COVID-19 [44]. However, these digital tools still need to be ensured to be scientifically and ethically sound to ensure widespread public trust and uptake [44]. Typological analysis and established public health and big data ethics frameworks can aid governments and other actors in discerning the complex ethical–legal landscape in which these digital tools will operate [45]. Moreover, the effectiveness of contact tracing and the extent of resources required to implement it successfully will ultimately depend on the social interactions [46]. A recent review suggests that COVID-19 contact tracing systems could be facilitated by: clear communication about contact tracing; involvement of stakeholders in the development of contact tracing systems, particularly digital applications; evaluation and quality assurance of the contact tracing system [34].

The key strength of this work was the ability to achieve consensus relating to the government's response to an ongoing pandemic from a considerable number of participants from multiple geographical regions across Indonesia. The process was completed in the middle of a global pandemic over a relatively short period (six weeks), without the ability to hold face-to-face meetings.

Notwithstanding, our Delphi process had key limitations. Firstly, the scope of the work precluded the inclusion of academics beyond medicine and health (e.g. economics, public policy, public administration) and non-academics (e.g. public health practitioners, clinical practitioners, policymakers, community leaders) in the process. Seeking opinions from a broader category of participants would have been preferable had time and the need to ensure participants' expertise not been such pressing factors. Secondly, the statements assessed here were all developed based on expert opinions [5]. As the output and recommendations from a Delphi process can only be as robust as the assessed statements, higher levels of evidence and further experiences should be included as they become available to ensure that the recommendations remain relevant.

## Conclusions

The COVID-19 pandemic has caused a global crisis that endures for the long haul [47]. Pressure on governments to act decisively is enormous. Citizens look to governments for

leadership and credible information [48]. To revamp the COVID-19 response in Indonesia, the government must first strengthen capacity to ensure consistent, credible and targeted communication while adopting a more inclusive and empathic communication style to address public concerns. Second, the Indonesian government must ensure universal access to reliable COVID-19 test by expanding lab infrastructure and providing universal access to reliable COVID-19 test by expanding lab infrastructure, facilitating operational readiness, and scaling up implementation of proven alternative/complementary tests to RT-PCR. Finally, the government need to boost contact tracing implementation capacity and facilitate contact tracing for all positive cases, involving key stakeholders in further development of the existing contact tracing system (i.e. PeduliLindungi) as well as its evaluation and quality assurance.

The recommendations from this Delphi study were primarily intended for the Indonesian government. Notwithstanding, governments of other countries may also benefit from this study by considering conducting a similar exercise, utilizing the modified Delphi method for rapidly assessing their response to COVID-19. Ultimately, our study highlights the importance of strengthening health system functions during the pandemic and improving health system resilience for dealing with future public health emergencies.

## Supporting information

**S1 File. CREDES checklist.**
(DOCX)

**S2 File. Delphi study data.**
(XLSX)

**S3 File. Delphi study code book.**
(DOCX)

## Acknowledgments

We would like to thank Ari Probandari, Bagoes Widjanarko, Chatarina Umbul Wahyuni, Dewi Susanna, I Wayan Gede Artawan Eka Putra, Mohammad Bakhriansyah, Rahayu Lubis, Sukri Palutturi, and Trevino Pakasi for their contribution to identify and recommend the potential participants for the modified Delphi study.

## Author Contributions

**Conceptualization:** Yodi Mahendradhata, Trisasi Lestari, Riyanti Djalante.

**Formal analysis:** Trisasi Lestari.

**Investigation:** Yodi Mahendradhata, Trisasi Lestari.

**Methodology:** Yodi Mahendradhata, Trisasi Lestari.

**Supervision:** Riyanti Djalante.

**Writing – original draft:** Yodi Mahendradhata, Trisasi Lestari.

**Writing – review & editing:** Riyanti Djalante.

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
