## [Decision Letter · Decision Letter 0]

17 May 2021

PONE-D-20-34947

Strengthening government’s response to COVID-19 in Indonesia: a modified Delphi study of medical and health academics

PLOS ONE

Dear Dr. Mahendradhata,

Thank you for submitting your manuscript to PLOS ONE. After careful consideration, we feel that it has merit but does not fully meet PLOS ONE’s publication criteria as it currently stands. Therefore, we invite you to submit a revised version of the manuscript that addresses the points raised during the review process.

We look forward to receiving your revised manuscript.

Kind regards,

Rubeena Zakar, Ph.D

Academic Editor

PLOS ONE

Additional Editor Comments:

Please strictly follow authors' guidelines by the journal while revising your manuscript. 

Please find below reviewers' comments. 

Journal Requirements:

3. In the Methods section, please describe in detail by what method the webinar discussion sessions were conducted. For instance, please provide information on the duration of the webinar, who conducted the webinar and their training experience and background, duration and finally how content was recorded and analysed.

Reviewers' comments:

Reviewer's Responses to Questions

**Comments to the Author**

1. Is the manuscript technically sound, and do the data support the conclusions?

Reviewer #1: Partly

Reviewer #2: Yes

2. Has the statistical analysis been performed appropriately and rigorously? 

Reviewer #1: Yes

Reviewer #2: N/A

3. Have the authors made all data underlying the findings in their manuscript fully available?

Reviewer #1: Yes

Reviewer #2: No

4. Is the manuscript presented in an intelligible fashion and written in standard English?

Reviewer #1: Yes

Reviewer #2: Yes

5. Review Comments to the Author

Reviewer #1: The authors aim to evaluate and provide further direction to Indonesia's Covid-19 response. To do so, they rely on a Delphi-style survey using the COVID-SCORE tool. Ultimately, this allows them to evaluate the government's response and identify key areas for improvement. This is clearly an important topic that merits consideration.

Sadly, in the current form, I cannot recommend publication. I will not comment on individual aspects of the manuscript but rather I wish to focus on what I see as the two related major hindrances: the methodology and the discussion/recommendations.

The methodology is very rigid and largely seems to be a the same questionnaire applied twice. Though there are optional short responses, there seems to be little room to obtain more in-depth observations and recommendations from the experts. An open-ended survey or focus group discussion with selected experts would be needed to do so. Though you mention having held a webinar, it is not clear if this helped build on the results in any way.

This leads to the second major issue. The recommendations in the manuscript are vague and therefore of limited use. For example, you mention the importance of a testing strategy or contact tracing. Yet there is little specific insight or recommendations on how these items are being specifically implemented in Indonesia and how they can be improved. In other words, the manuscript merely emphasizes the importance of such items, but there is little country-specific information provided.

Moving forward, more specific recommendations and context need to be provided. This can be done by sourcing additional local news/policy/literature as well as by conducting a third qualitative step in your Delphi study, whereby you validate and discuss the results in one or more focus groups with relevant experts. I feel such a step is needed to deepen the manuscript and maximize its relevance.

Reviewer #2: The paper is important to inform the Indonesia government in tackling the COVID-19 pandemic. It is well written in very good English and understandable to read. The strength of this paper is the implementation of the COVID-19 assessment scorecard (COVID-SCORE) in a modified Delphi method. The results may be used to develop new models of preparedness and response for future public-health threads.

To make it better, the authors may:

• Line 156 � please clarify that the second cycle is the final results (no other returning the questionnaire to the researchers again)

• Line 189 � it may be better if the majority comes first (68.5% doctoral degree vs 31.5% master degree)

• Line 192-193 � how about the 17 participants who directly involved with COVID-19 response team? Why were they not excluded from the survey related to their possibility of subjectivity?

• Line 207/ Table 1. There is a column of “Neither agree nor disagree” and the columns for assessment of comments consisting of positive, negative, or mixed comments. However, in the analysis, only ‘agreement’ or ‘disagreement’ are described. Also, there are no definitions for positive, negative, and mixed sentiment.

• Line 468. After stating the Wuhan government failure in early communication, it will be more informative if there are any data about communication of Indonesia government in the early pandemic, such as the early response of the previous Minister of Health who was not able to build the public trusts. Also, the role of MoH spokesman should be discussed.

• Line 487-488. The important of social media in communication � is there any data about the quality of government’s social media and how the government tackle the false pandemic information. There are also a few independent organizations that manage social media account to provide pandemic information and also to identify discrepancy data in COVID-19 daily report (e.g., @pandemictalks; @kawalcovid19) � it would be interesting to discuss their role in public health communication and how the government response.

• Line 494-510 about enhancing testing and contact tracing � it would be useful if the authors update the impacts of latest diagnostic methods used in Indonesia (e.g., RDT antigen and Genose) as these methods have already been regulated by the government to trace and screen.

• P 522-523, about app-based digital tracing � there is an interesting article to enrich the discussion: Gasser U, Ienca M, Scheibner J, Sleigh J, Vayena E. Digital tools against COVID-19: taxonomy, ethical challenges, and navigation aid. Lancet Digit Health. 2020 Aug;2(8):e425-e434. doi: 10.1016/S2589-7500(20)30137-0. Epub 2020 Jun 29. PMID: 32835200; PMCID: PMC7324107.

• Related to app-based digital tracing, kindly update the Indonesian government-sponsored tracing app, PeduliLindungi. PeduliLindungi is an android-based mobile application developed by the Indonesian government to mitigate the spread of COVID-19. The application relies on the willingness of people to share their current health condition and make it available for tracking using global positioning system (GPS) and Bluetooth technology.

• In the analysis section, the authors mentioned the webinar between experts and the National Planning Development Agency to discuss findings of the study� it would be nice if authors discuss the responses from the National Planning Development Agency during the webinar and provide the outcomes of the meeting.

6. PLOS authors have the option to publish the peer review history of their article (what does this mean?). If published, this will include your full peer review and any attached files.

Reviewer #1: **Yes: **Louis Moustakas

Reviewer #2: No

---

## [Author Response · Author response to Decision Letter 0]

20 Jul 2021

Dear Editor, many thanks for the comments. Please find below point-by-point responses to the comments.

We have reviewed the style requirements and adjusted the manuscript accordingly.

No minors were involved in our study, and we obtained written informed consent from all participants. We've added the following text to the ethic sub-section of the method section: 

"Written informed consent was required before a participant could start filling the online questionnaire."

3. In the Methods section, please describe in detail by what method the webinar discussion sessions were conducted. For instance, please provide information on the duration of the webinar, who conducted the webinar and their training experience and background, duration and finally how content was recorded and analysed.

We have added the following text in the method section:

"The webinar was delivered via Zoom, facilitated by the authors and completed in 1,5 hours. During the webinar, respondents were asked whether they agree with the presented results and priority ranking. The authors documented their responses and the entire webinar was audio and video recorded."

Reviewer #1: The authors aim to evaluate and provide further direction to Indonesia's Covid-19 response. To do so, they rely on a Delphi-style survey using the COVID-SCORE tool. Ultimately, this allows them to evaluate the government's response and identify key areas for improvement. This is clearly an important topic that merits consideration. Sadly, in the current form, I cannot recommend publication. I will not comment on individual aspects of the manuscript but rather I wish to focus on what I see as the two related major hindrances: the methodology and the discussion/recommendations.

Thank you for highlighting the importance of the topic. We address below the two hindrances noted by the reviewer

The methodology is very rigid and largely seems to be a the same questionnaire applied twice. Though there are optional short responses, there seems to be little room to obtain more in-depth observations and recommendations from the experts. 

We concur that there are limitations to the Delphi technique. Notwithstanding, this technique is a well-established approach to answering a research question by identifying a consensus view across subject experts. The Delphi technique has been used internationally to investigate a wide variety of issues, including health issues. This technique is primarily used in health sciences by researchers when the available knowledge is incomplete or subject to uncertainty, and other methods that provide higher levels of evidence cannot be used. Thus, we believe applying the Delphi method rigorously can provide answers which are useful and credible. We have added the following text in the method section to highlight this

" The Delphi technique is a well-established approach to answering a research question by identifying a consensus view across subject experts [11]. This technique is commonly used in developing consensual guidance on best practice and exploration of a field beyond existing knowledge and the current conceptual framework [12]. The Delphi technique has been used internationally to investigate a wide variety of issues, including health issues. This technique is primarily used in health sciences by researchers when the available knowledge is incomplete or subject to uncertainty, and other methods that provide higher levels of evidence cannot be used [13]."

An open-ended survey or focus group discussion with selected experts would be needed to do so. Though you mention having held a webinar, it is not clear if this helped build on the results in any way.

We agree that a follow-up survey and FGDs would be useful. These have been carried out subsequently and has led to a separate manuscript which has been published elsewhere. We have added the following text in the result section to clarify this:

" During the webinar to discuss these results, experts confirmed their agreement with the presented results and priority ranking. They also suggested recommendations for follow-ups of the study. Accordingly, the results were shared with the National Planning Development Agency and informed the development of a follow-up study. Some of the results of this subsequent study has now been published elsewhere [16]."

This leads to the second major issue. The recommendations in the manuscript are vague and therefore of limited use. For example, you mention the importance of a testing strategy or contact tracing. Yet there is little specific insight or recommendations on how these items are being specifically implemented in Indonesia and how they can be improved. In other words, the manuscript merely emphasizes the importance of such items, but there is little country-specific information provided. Moving forward, more specific recommendations and context need to be provided. This can be done by sourcing additional local news/policy/literature as well as by conducting a third qualitative step in your Delphi study, whereby you validate and discuss the results in one or more focus groups with relevant experts. I feel such a step is needed to deepen the manuscript and maximize its relevance.

We have added the following texts in the discussion section to provide additional country specific context as requested:

 " The only source of official information at the early phase of the Pandemic in Indonesia came from the presidential office and the Indonesian Health Ministry. The government did not appear to have a "sense of crisis" even as of early February 2020. This was exacerbated by "antiscience" statements issued by government officials [19]. The government appointed a spokesman who informed the progress of the spread of COVID-19 through one door regularly and periodically twice a day. However, when the number of cases increased in several areas, mayors and governors also issued statements, which often contradict the official data held by the presidential office and Minister of Health [20]."

"The Indonesian government has implemented government-centered public communication without prohibiting media, including social media, from spreading general information about COVID-19 [25]. Thus, the primary sources of information for the public during the pandemic are social media, online media, and mass media. The most popular sources for preventive measures are social media (83.6%), television (78.5%), and WhatsApp (76.0%) [26]. Notably, Indonesia faces problems with the spread of misleading and harmful messages that hamper COVID-19 prevention efforts. The Ministry of Communications and Informatics monitors the spread of COVID-19 related mis/dis-information daily, with records of 1,550 hoaxes related to Covid-19 dan 177 hoaxes related to Covid-19 vaccines from January 2020 to April 2021 [27]. In response, the government has partnered with digital technology-based industries and social media influencers for disseminating information [28]."

"The gap between the RT-PCR testing capacity in laboratories and the number of suspected cases to be tested remains significant. Accordingly, the government adopted the WHO recommendation to use antigen detection rapid diagnostic test (Ag-RDT) to diagnose SARS-CoV-2 infection in settings where RT-PCR is unavailable or where prolonged turnaround precludes clinical utility [33]. The utilization of Ag-RDT has augmented the country's testing capacity. More recently, the government has also adopted GeNose C19, a breath-based COVID-19 detection device developed by Indonesian scientists. Initial trial on the device revealed 89-92% sensitivity for detecting positive cases, comparable to RT-PCR (89%) and antigen-based rapid test (89.9%) [34]. The tool that detects the infection through exhaled volatile organic compounds gives results in only three minutes. It costs only US$ 0.7-1.7 or around 11-140 times cheaper than the other two tests. [34] The Ministry of Transportation has deployed the tool in train stations and bus terminals, hoping that the device can speed up the passengers screening process."

"The Ministry of Information and Communication launched a mobile application called PeduliLindungi. The app enables users to compile data related to the spread of COVID-19 in their communities to help bolster the Indonesian government's efforts to trace and track confirmed cases. Users are expected to register as participants and share their locations when travelling and trace whether they have contacted persons exposed to COVID-19. The app will also alert users entering crowds or COVID-19 red zones, namely areas with confirmed COVID-19 cases [39]."

We have also added the following text to the conclusion section

"To revamp the COVID-19 response in Indonesia, the government first must urgently need to strengthen capacity to ensure consistent, credible and targeted communication while adopting a more inclusive and empathic communication style to address public concerns. Second, the Indonesian government need to ensure universal access to reliable COVID-19 test by expanding lab infrastructure and to ensure universal access to reliable COVID-19 test by expanding lab infrastructure, facilitating operational readiness, and scaling up implementation of proven alternative/complementary tests to RT-PCR. Finally, the government need to boost contact tracing implementation capacity and facilitate contact tracing for all positive cases, involving key stakeholders in further development of the existing contact tracing system (i.e. PeduliLindungi) as well as its evaluation and quality assurance."

Reviewer #2: The paper is important to inform the Indonesia government in tackling the COVID-19 Pandemic. It is well written in very good English and understandable to read. The strength of this paper is the implementation of the COVID-19 assessment scorecard (COVID-SCORE) in a modified Delphi method. The results may be used to develop new models of preparedness and response for future public-health threads.

To make it better, the authors may:

• Line 156 � please clarify that the second cycle is the final results (no other returning the questionnaire to the researchers again)

We have revised the sentence as follow to emphasize that the second cycle was the final cycle:

"The result of the second (final) cycle was analyzed, and feedback to all participants a week later."

• Line 189 � it may be better if the majority comes first (68.5% doctoral degree vs 31.5% master degree)

We have revised as follow in line with the suggestion:

"Most of the participants hold a doctoral degree (48, 68.5%) and the rest hold a master degree (22, 31.5%)."

• Line 192-193 � how about the 17 participants who directly involved with COVID-19 response team? Why were they not excluded from the survey related to their possibility of subjectivity?

The 17 participants who were directly involved with the COVID-19 response were not excluded as we consider their involvement to be added value in providing insights as those who were close with the realities of COVID-19 response in the field.

• Line 207/ Table 1. There is a column of "Neither agree nor disagree" and the columns for assessment of comments consisting of positive, negative, or mixed comments. However, in the analysis, only 'agreement' or 'disagreement' are described. Also, there are no definitions for positive, negative, and mixed sentiment.

We have now deleted the positive, negative or mixed comments as the focus of our analysis is on the agreement/disagreement and priority ranking

• Line 468. After stating the Wuhan government failure in early communication, it will be more informative if there are any data about communication of Indonesia government in the early Pandemic, such as the early response of the previous Minister of Health who was not able to build the public trusts. Also, the role of MoH spokesman should be discussed.

We have added the following text to the discussion section as advised:

"The only source of official information at the early phase of the Pandemic in Indonesia came from the presidential office and the Indonesian Health Ministry. The government did not appear to have a "sense of crisis" even as of early February 2020. This was exacerbated by "antiscience" statements issued by government officials [19]. The government appointed a spokesman who informed the progress of the spread of COVID-19 through one door regularly and periodically twice a day. However, when the number of cases increased in several areas, mayors and governors also issued statements, which often contradict the official data held by the presidential office and Minister of Health [20]."

• Line 487-488. The important of social media in communication � is there any data about the quality of government's social media and how the government tackle the false pandemic information. There are also a few independent organizations that manage social media account to provide pandemic information and also to identify discrepancy data in COVID-19 daily report (e.g., @pandemictalks; @kawalcovid19) � it would be interesting to discuss their role in public health communication and how the government response.

"The Indonesian government has implemented government-centered public communication without prohibiting media, including social media, from spreading general information about COVID-19 [25]. Thus, the primary sources of information for the public during the pandemic are social media, online media, and mass media. The most popular sources for preventive measures are social media (83.6%), television (78.5%), and WhatsApp (76.0%) [26]. Notably, Indonesia faces problems with the spread of misleading and harmful messages that hamper COVID-19 prevention efforts. The Ministry of Communications and Informatics monitors the spread of COVID-19 related mis/dis-information daily, with records of 1,550 hoaxes related to Covid-19 dan 177 hoaxes related to Covid-19 vaccines from January 2020 to April 2021 [27]. In response, the government has partnered with digital technology-based industries and social media influencers for disseminating information [28]."

• Line 494-510 about enhancing testing and contact tracing � it would be useful if the authors update the impacts of latest diagnostic methods used in Indonesia (e.g., RDT antigen and Genose) as these methods have already been regulated by the government to trace and screen."

We have added the following text to the discussion section as advised:

"The gap between the RT-PCR testing capacity in laboratories and the number of suspected cases to be tested remains significant. Accordingly, the government adopted the WHO recommendation to use antigen detection rapid diagnostic test (Ag-RDT) to diagnose SARS-CoV-2 infection in settings where RT-PCR is unavailable or where prolonged turnaround precludes clinical utility [33]. The utilization of Ag-RDT has augmented the country's testing capacity. More recently, the government has also adopted GeNose C19, a breath-based COVID-19 detection device developed by Indonesian scientists. Initial trial on the device revealed 89-92% sensitivity for detecting positive cases, comparable to RT-PCR (89%) and antigen-based rapid test (89.9%) [34]. The tool that detects the infection through exhaled volatile organic compounds gives results in only three minutes. It costs only US$ 0.7-1.7 or around 11-140 times cheaper than the other two tests. [34] The Ministry of Transportation has deployed the tool in train stations and bus terminals, hoping that the device can speed up the passengers screening process."

• P 522-523, about app-based digital tracing � there is an interesting article to enrich the discussion: Gasser U, Ienca M, Scheibner J, Sleigh J, Vayena E. Digital tools against COVID-19: taxonomy, ethical challenges, and navigation aid. Lancet Digit Health. 2020 Aug;2(8):e425-e434. doi: 10.1016/S2589-7500(20)30137-0. Epub 2020 Jun 29. PMID: 32835200; PMCID: PMC7324107.

We have incorporated the recommended reference into the discussion section as well:

"However, these digital tools still need to be ensured to be scientifically and ethically sound to ensure widespread public trust and uptake [40]. Typological analysis and established frameworks in public health and big data ethics can aid governments and other actors in discerning the complex ethical–legal landscape in which these digital tools will operate [40]. "

• Related to app-based digital tracing, kindly update the Indonesian government-sponsored tracing app, PeduliLindungi. PeduliLindungi is an android-based mobile application developed by the Indonesian government to mitigate the spread of COVID-19. The application relies on the willingness of people to share their current health condition and make it available for tracking using global positioning system (GPS) and Bluetooth technology.

We have added the following text to the discussion section as advised:

"The Ministry of Information and Communication launched a mobile application called PeduliLindungi. The app enables users to compile data related to the spread of COVID-19 in their communities to help bolster the Indonesian government's efforts to trace and track confirmed cases. Users are expected to register as participants and share their locations when travelling and trace whether they have contacted persons exposed to COVID-19. The app will also alert users entering crowds or COVID-19 red zones, namely areas with confirmed COVID-19 cases [39]. "

• In the analysis section, the authors mentioned the webinar between experts and the National Planning Development Agency to discuss findings of the study� it would be nice if authors discuss the responses from the National Planning Development Agency during the webinar and provide the outcomes of the meeting.

We have added the following text to the result section as advised:

"During the webinar to discuss these results, experts confirmed their agreement with the presented results and priority ranking. They also suggested recommendations for follow-ups of the study. Accordingly, the results were shared with the National Planning Development Agency and informed the development of a follow-up study. Some of the results of this subsequent study has now been published elsewhere [16]."

---

## [Decision Letter · Decision Letter 1]

20 Sep 2021

PONE-D-20-34947R1Strengthening government’s response to COVID-19 in Indonesia: a modified Delphi study of medical and health academicsPLOS ONE

Dear Dr. Mahendradhata,

Thank you for submitting your manuscript to PLOS ONE. After careful consideration, we feel that it has merit but does not fully meet PLOS ONE’s publication criteria as it currently stands. Therefore, we invite you to submit a revised version of the manuscript that addresses the points raised during the review process. Please submit your revised manuscript by 30th October 2021. If you will need more time than this to complete your revisions, please reply to this message or contact the journal office at plosone@plos.org. Please include the following items when submitting your revised manuscript:A rebuttal letter that responds to each point raised by the academic editor and reviewer(s). You should upload this letter as a separate file labeled 'Response to Reviewers'.A marked-up copy of your manuscript that highlights changes made to the original version. You should upload this as a separate file labeled 'Revised Manuscript with Track Changes'.An unmarked version of your revised paper without tracked changes. You should upload this as a separate file labeled 'Manuscript'.If applicable, we recommend that you deposit your laboratory protocols in protocols.io to enhance the reproducibility of your results. Protocols.io assigns your protocol its own identifier (DOI) so that it can be cited independently in the future. For instructions see: https://journals.plos.org/plosone/s/submission-guidelines#loc-laboratory-protocols. Additionally, PLOS ONE offers an option for publishing peer-reviewed Lab Protocol articles, which describe protocols hosted on protocols.io. Read more information on sharing protocols at https://plos.org/protocols?utm_medium=editorial-email&utm_source=authorletters&utm_campaign=protocols.

We look forward to receiving your revised manuscript.

Kind regards,

Rubeena Zakar, Ph.D

Academic Editor

PLOS ONE

Journal Requirements:

Reviewers' comments:

Reviewer's Responses to Questions

**Comments to the Author**

1. If the authors have adequately addressed your comments raised in a previous round of review and you feel that this manuscript is now acceptable for publication, you may indicate that here to bypass the “Comments to the Author” section, enter your conflict of interest statement in the “Confidential to Editor” section, and submit your "Accept" recommendation.

Reviewer #2: (No Response)

Reviewer #3: (No Response)

2. Is the manuscript technically sound, and do the data support the conclusions?

Reviewer #2: Yes

Reviewer #3: Yes

3. Has the statistical analysis been performed appropriately and rigorously? 

Reviewer #2: Yes

Reviewer #3: Yes

4. Have the authors made all data underlying the findings in their manuscript fully available?

Reviewer #2: Yes

Reviewer #3: Yes

5. Is the manuscript presented in an intelligible fashion and written in standard English?

Reviewer #2: Yes

Reviewer #3: Yes

6. Review Comments to the Author

Reviewer #2: Thank you for addressing my issues. I would like to congratulate you on the improvement in your paper. I only have one issue left regarding the GeNose C19 since there was updating news.

You added:

“More recently, the government has also adopted GeNose C19, a breath-based COVID-19 detection device developed by Indonesian scientists. Initial trial on the device revealed 89-92% sensitivity for detecting positive cases, comparable to RT-PCR (89%) and antigen-based rapid test (89.9%) [34]. The tool that detects the infection through exhaled volatile organic compounds gives results in only three minutes. It costs only US$ 0.7-1.7 or around 11-140 times cheaper than the other two tests. [34] The Ministry of Transportation has deployed the tool in train stations and bus terminals, hoping that the device can speed up the passengers screening process."

As we know that the GeNose C19 is now halted by the Government upon its controversy. I would like you to delete the discussion about the GeNose C19 to avoid the confusion. Kindly highlight the effort from the Government in providing effective and reliable COVID-19 screening modalities, and also how the Government take the expert and epidemiologist recommendation. Thank you.

Reviewer #3: Research Summary and overall impression

The study assessed the response efforts by the Indonesian government to the COVID-19 pandemic from the time of first outbreak in February, 2020 in Indonesia. This work involved medical and health academics from ten universities across Indonesia, whose views on 20 prepared statements were captured using a five-point Likert scale in a modified Delphi study. Suggested areas requiring improvement were also put forth by participants.

Overall, this was a good study, which has been well-presented. This is a relevant study as far as COVID-19-related Public Health issues are concerned. Some insight is provided the reader into the strengths and shortfalls of the Indonesian government in its handling of the COVID-19 outbreak in the archipelagos – issues which are common to many countries across the globe, developed and developing alike.

There however are a few observations which the authors should note:

Major Issues

1. Line 369: (majority agreement, 51.4% disagree) – this is a rather confusing presentation. Although one deduces from reading further that the first agreement is actually degree of consensus, and the second agreement with percentage indicates the kind of agreement (whether agreed or disagreed), it could adversely impact the smooth reading and flow of the manuscript.

I would suggest that different categorizations (such as ‘poor consensus’, ‘consensus’, ‘good consensus’ and ‘strong consensus’) could be adopted instead of what is currently being used (i.e. ‘no consensus’, ‘majority agreement’, ‘consensus’, and ‘strong consensus’). This could also greatly improve understanding at first read. Same issue applies to Table 1 as well

2. Table 1: I’m not so sure what the relevance of the p-value column is in this work. No mention is made of it in either the results or the discussion sections. Meanwhile, this could serve to justify why it was needful to conduct the study twice with the same participants. I think it would enrich the manuscript if they could insert such information

Minor Issues

1. Lines 439 – 440: I think the authors should be more careful with their tenses. It would be helpful if they could go through the entire manuscript to correct these issues.

2. Lines 389 – 392: Unclear statement. Please revise.

3. Lines 518 - 519: The clause ‘through social media and employ strategies for improved communication management to citizens through social media…’ was repeated. Please revise.

Other/General remarks:

I think the study was well-done and well-presented. However, attention must be paid to modify the categorizations; as well as correct tenses and other minor grammatical errors. This would greatly enhance the readability of the study.

I would recommend it for publication. Thank you.

7. PLOS authors have the option to publish the peer review history of their article (what does this mean?). If published, this will include your full peer review and any attached files.

Reviewer #2: No

Reviewer #3: No

---

## [Author Response · Author response to Decision Letter 1]

26 Sep 2021

Dear Editor,

Thank you for the reviewers' comments. We've now revised the manuscript in line with the comments. Please find below point-by-point responses to the comments. 

Best regards,

Yodi Mahendradhata

Reviewer #2: Thank you for addressing my issues. I would like to congratulate you on the improvement in your paper. I only have one issue left regarding the GeNose C19 since there was updating news.

You added:

"More recently, the government has also adopted GeNose C19, a breath-based COVID-19 detection device developed by Indonesian scientists. Initial trial on the device revealed 89-92% sensitivity for detecting positive cases, comparable to RT-PCR (89%) and antigen-based rapid test (89.9%) [34]. The tool that detects the infection through exhaled volatile organic compounds gives results in only three minutes. It costs only US$ 0.7-1.7 or around 11-140 times cheaper than the other two tests. [34] The Ministry of Transportation has deployed the tool in train stations and bus terminals, hoping that the device can speed up the passengers screening process."

As we know that the GeNose C19 is now halted by the Government upon its controversy. I would like you to delete the discussion about the GeNose C19 to avoid the confusion. Kindly highlight the effort from the Government in providing effective and reliable COVID-19 screening modalities, and also how the Government take the expert and epidemiologist recommendation. Thank you.

The text on GeNose C19 in the discussion section has been deleted and replaced with text to highlight government effort in reaching WHO recommended testing rate and enhancing capacity for detecting variants of concerns as follow:

" As of September 19 2021, all provinces in Indonesia have managed to meet the WHO recommended testing rate of at least one person per 1000 population per week [34]. Additionally, to enhance capacity to detect variants of concerns, the Ministry of Health has incorporated 21 institutions into its SARS-CoV-2 genome surveillance laboratory network, with the capacity of around 1,750 – 2,700 tests per month. As of August 29 2021, Indonesia has shared 5,755 (0.142% cases) sequences since the beginning of the pandemic, according to GISAID (Global Initiative on sharing all Influenza data). By August 2021, the Ministry of Health announced that the Delta Variant had been detected in six provinces, i.e. DKI Jakarta, West Java, East Kalimantan, Central Java, East Nusa Tenggara, and Bali 1 2021 [35]." 

Reviewer #3: Research Summary and overall impression

The study assessed the response efforts by the Indonesian government to the COVID-19 pandemic from the time of first outbreak in February, 2020 in Indonesia. This work involved medical and health academics from ten universities across Indonesia, whose views on 20 prepared statements were captured using a five-point Likert scale in a modified Delphi study. Suggested areas requiring improvement were also put forth by participants.

Overall, this was a good study, which has been well-presented. This is a relevant study as far as COVID-19-related Public Health issues are concerned. Some insight is provided the reader into the strengths and shortfalls of the Indonesian government in its handling of the COVID-19 outbreak in the archipelagos – issues which are common to many countries across the globe, developed and developing alike.

There however are a few observations which the authors should note:

Major Issues

1. Line 369: (majority agreement, 51.4% disagree) – this is a rather confusing presentation. Although one deduces from reading further that the first agreement is actually degree of consensus, and the second agreement with percentage indicates the kind of agreement (whether agreed or disagreed), it could adversely impact the smooth reading and flow of the manuscript.

I would suggest that different categorizations (such as 'poor consensus', 'consensus', 'good consensus' and 'strong consensus') could be adopted instead of what is currently being used (i.e. 'no consensus', 'majority agreement', 'consensus', and 'strong consensus'). This could also greatly improve understanding at first read. Same issue applies to Table 1 as well

We concur that the term "majority agreement" may lead to confusion. Thus, the categorization has now been modified to "no consensus", "low consensus level", "moderate consensus level" and "high consensus level." This modification follows categorization described by Meskell et al (2014). The modified categorization has been described in the method section as follow:

"High consensus level" was defined as ≥95% agreement in the rating of the single statements by the panellists; "Moderate consensus level" was defined as the agreement of 75–94%, "Low consensus level" was defined as the agreement of 50-74%, and "No consensus" was recorded if the agreement was <50%."

The new categorization has been applied to Table one and the relevant texts in the result dan discussion sections.

2. Table 1: I'm not so sure what the relevance of the p-value column is in this work. No mention is made of it in either the results or the discussion sections. Meanwhile, this could serve to justify why it was needful to conduct the study twice with the same participants. I think it would enrich the manuscript if they could insert such information

The following additional texts have now been inserted into the method section:

" In the second cycle, respondents had the opportunity to revise their answers. We calculated p-values to assess the significance of the change in the degree of consensus for each statement between the cycles."

And the result section:

" Compared to the first cycle, the consensus degree was improved for 15 statements and reduced for four statements. However, a significant change (p < 0.05) of consensus degree between the cycles was only observed for one statement ("Mental health outreach services have been expanded to meet increased demand")."

Minor Issues

1. Lines 439 – 440: I think the authors should be more careful with their tenses. It would be helpful if they could go through the entire manuscript to correct these issues.

The noted text has been revised as follow, and the rest of the manuscript has been thoroughly checked using Grammarly Business :

" The government have released financial incentives for the poor and encouraged older adults and young children to stay at home."

2. Lines 389 – 392: Unclear statement. Please revise.

The noted text has been revised as follow:

" Respondents suggest campaigns should adapt local wisdom and use language that should be easy for people with low literacy to understand. The use of the internet and social media to share information has discriminated population living in rural without access to the internet or television."

3. Lines 518 - 519: The clause 'through social media and employ strategies for improved communication management to citizens through social media…' was repeated. Please revise.

The noted text has been revised as follow:

" The Indonesian government thus need to improve their efforts to disseminate information on the pandemic and employ strategies for improved communication management to citizens through social media and mainstream information sources [24]."

Other/General remarks:

I think the study was well-done and well-presented. However, attention must be paid to modify the categorizations; as well as correct tenses and other minor grammatical errors. This would greatly enhance the readability of the study.

I would recommend it for publication. Thank you.

The categorization has now been modified as described above, and the manuscript has now been rechecked for grammatical errors using Grammarly Business.

---

## [Decision Letter · Decision Letter 2]

10 Jan 2022

PONE-D-20-34947R2Strengthening government’s response to COVID-19 in Indonesia: a modified Delphi study of medical and health academicsPLOS ONE

Dear Dr. Mahendradhata,

Thank you for submitting your manuscript to PLOS ONE. After careful consideration, we feel that it has merit but does not fully meet PLOS ONE’s publication criteria as it currently stands. Therefore, we invite you to submit a revised version of the manuscript that addresses the points raised during the review process.

Please submit your revised manuscript by 24 February 2022. If you will need more time than this to complete your revisions, please reply to this message or contact the journal office at plosone@plos.org. Please include the following items when submitting your revised manuscript:A rebuttal letter that responds to each point raised by the academic editor and reviewer(s). You should upload this letter as a separate file labeled 'Response to Reviewers'.A marked-up copy of your manuscript that highlights changes made to the original version. You should upload this as a separate file labeled 'Revised Manuscript with Track Changes'.An unmarked version of your revised paper without tracked changes. You should upload this as a separate file labeled 'Manuscript'.If applicable, we recommend that you deposit your laboratory protocols in protocols.io to enhance the reproducibility of your results. Protocols.io assigns your protocol its own identifier (DOI) so that it can be cited independently in the future. For instructions see: https://journals.plos.org/plosone/s/submission-guidelines#loc-laboratory-protocols. Additionally, PLOS ONE offers an option for publishing peer-reviewed Lab Protocol articles, which describe protocols hosted on protocols.io. Read more information on sharing protocols at https://plos.org/protocols?utm_medium=editorial-email&utm_source=authorletters&utm_campaign=protocols.

We look forward to receiving your revised manuscript.

Kind regards,

Rubeena Zakar, Ph.D

Academic Editor

PLOS ONE

Journal Requirements:

Reviewers' comments:

Reviewer's Responses to Questions

**Comments to the Author**

1. If the authors have adequately addressed your comments raised in a previous round of review and you feel that this manuscript is now acceptable for publication, you may indicate that here to bypass the “Comments to the Author” section, enter your conflict of interest statement in the “Confidential to Editor” section, and submit your "Accept" recommendation.

Reviewer #3: All comments have been addressed

Reviewer #4: All comments have been addressed

Reviewer #5: (No Response)

Reviewer #6: (No Response)

2. Is the manuscript technically sound, and do the data support the conclusions?

Reviewer #3: Yes

Reviewer #4: Yes

Reviewer #5: Partly

Reviewer #6: Yes

3. Has the statistical analysis been performed appropriately and rigorously? 

Reviewer #3: Yes

Reviewer #4: Yes

Reviewer #5: No

Reviewer #6: Yes

4. Have the authors made all data underlying the findings in their manuscript fully available?

Reviewer #3: Yes

Reviewer #4: Yes

Reviewer #5: Yes

Reviewer #6: Yes

5. Is the manuscript presented in an intelligible fashion and written in standard English?

Reviewer #3: Yes

Reviewer #4: Yes

Reviewer #5: Yes

Reviewer #6: Yes

6. Review Comments to the Author

Reviewer #3: Overall Impressions:

All major issues have been catered for by the co-authors. There are however a number of minor issues, mainly typographical and grammatical, that the co-authors would need to address:

Issues:

Location Issue

Line 19 ‘government’s’ was repeated twice

Line 19 Remove ‘the’ in ‘…government’s handling of the COVID-19…’

Line 25 Remove ‘have’ in the statement: ‘All participants who have completed…’

Line 27 ‘rank’ should be ‘ranked’

Replace ‘priority with ‘list’

Line 31 Replace ‘strengthen’ with ‘strengthening’

Line 33 Replace ‘ensure’ with ‘ensuring’

Line 35 Replace ‘boost’ with ‘boosting’

Line 36 Replace ‘facilitate’ with ‘facilitating’

Line 122 Replace ‘has’ with ‘had’

Line 132 Replace ‘university’ with ‘respective universities’

Line 176 Replace ‘will be’ with ‘was’

Line 177 Replace ‘will be’ with ‘was considered’

Line 188 Replace ‘can’ with ‘could’

Line 190 Replace ‘are’ with ‘were’

Line 225 Replace ‘has’ with ‘have’

Line 289 Capitalize ‘low’

Lines 290, 292, 298, 300, 301, 302, 304, 305 Replace ‘,’ with ‘;’

Line 300 Replace ‘communicate’ with ‘communicated’

Line 301 Replace ‘have’ with ‘has’

Line 303 Replace ‘has’ with ‘have’

Line 310 Replace ‘eight’ with ‘eighth’

Line 337 Modify ‘at what extent does’ to ‘the extent to which’

Line 359 Replace ‘needs’ with ‘needed’

Line 394 Replace ‘suggest’ with ‘suggested that’

Line 396 Modify ‘rural’ to read ‘rural areas’

Line 410 Replace ‘issue’ with ‘issues’

Line 417 Modify ‘price skyrocketing’ to read ‘prices were skyrocketing’

Line 446 Replace ‘philanthropic’ with ‘philanthropies’

Line 504 Replace ‘contradict’ with ‘contradicted

Line 531 Replace ‘dan’ with ‘and’

Reviewer #4: Line 15: Please make sure to use one common tense (either present or past) throughout the abstract. There are multiple grammatical mistakes in this paragraph. Please correct them.

Line: 19: Please omit the repeated words.

Line 23: Please clarify how many rounds of data collection was performed.

Line 46: Please be consistent when characterizing the COVID-19 as a pandemic. Here the authors have depicted it as an epidemic, which may confuse the readers.

Line 60: Please explain the term "zero case" - is it an academic term? If so, please provide a reference for it - otherwise please explain clearly for the readers.

Line 62: Should it be specific quarantines, or specific policies/instructions/guidelines for quarantines? Please clarify.

Line 64: Is there any data regarding Indonesian nationals versus tourists arriving in the country during that period of time? Please provide references as Indonesia is a known tourist country.

Line 67: Please provide information on the first COVID-19 related death occurring in Indonesia - when did it occur and how did that impact the governments' decisions?

Line 75: When the authors mention "Indonesia's vulnerable health system" - what do they mean? Please provide some context as to why the healthcare system is vulnerable in Indonesia.

Line 76: Please clarify what the non-COVID-19 related impacts are.

Line 78: Please be consistent with WHO announcements on COVID-19 being a pandemic.

Line 82: Please clarify the difference between health professionals (line 80) and health academics (line 82). Please use one term consistently throughout the paper.

Line 91: A better term that can be used here is "to achieve the study objectives".

Line 96: Please correct for double spaces throughout the manuscript. Please use single spaces.

Line 102: If possible, please provide references of other studies that have used a similar methodology to study the COVID-19 pandemic.

Line 104: Please clarify what the "modified Delphi study" is. Is this different from the original method? If so, how and why is it different? Please clarify for the readers.

Line 114: Please provide some context as to what type of "adopted statements" were utilized in the study. Are these statements from the Indonesian government? If so, which government body issued them and in what context? Was there a specific time frame from which the statements were collected? What was the inclusion criteria for these statements? Please clarify.

Line 126: Why is regional representation important in this context? Please clarify.

Line 127: When describing the sampling technique, please use more academic/formal terms such as purposive sampling etc.

Line 132: Did the authors follow a snowball sampling technique in the second stage of sampling? Please clarify the different stages that were used during the sampling process.

Line 134: Was the data collection process completed online? Please clarify this at the beginning of the methods section. Otherwise, it is confusing to the reader.

Line 137: How was this number (about 30) determined? Please clarify.

Line 148: Please clarify why 2 languages were used in the survey instrument.

Line 159: Why was not the data de-identified? Why did the researcher have access to the identity of the survey participants? Please clarify.

Line 163: Please clarify - was the feedback shared with the participants?

Line 167: Please correct the spelling for calculated.

Line 170: Please clarify how this decision criteria was determined. Please provide relevant references if possible.

Line 180: Please clarify what is meant by 1,5 hours.

Line 181: In case a respondent disagreed with the results - how was that handled by the researchers? What implications might it have for the results of the study?

Line 189: Earlier it was mentioned that the researchers/authors had access to participant identities. This could have affected the responses from the participants. How was this handled by the authors?

Line 287: Was this a recommendation made by the respondents?

Line 325: This can be better expressed as quality of product.

Line 337: Please correct grammatical issues with this sentence.

Line 344: Please provide a reference for this guideline.

Line 355: Please rewrite this sentence to make it clear for the readers.

Line 376: Please provide references for these resources.

Line 410: Please correct grammatical issues with this sentence.

Line 446: Please correct grammatical issues with this sentence.

Line 451: Are these 2 separate departments of the government? Please clarify.

Line 470: Overall, the discussion section should be made more concise, as several paragraphs can be summarized into smaller ones.

Line 471: The results mentioned here has already been mentioned in the results section. This paragraph is thus no longer needed.

Line 478: The results mentioned here has already been mentioned in the results section. This paragraph is thus no longer needed.

Line 513: Please summarize the discussion on communication in a more concise way.

Line 612: Please clarify the meaning of higher levels of evidence.

Reviewer #5: Thanks for the authors for introducing us how medical and health academics view the Indonesian government's handling of the COVID-19 and which areas of health systems need to be prioritized. However, there are several concerns:

1. Although the author explains the reasons for using the Delphi method in page 5 lines 93-95, this kind of research does not require the Delphi method (time-consuming and laborious). The same result would have been obtained if a questionnaire had been used. The commonly used Delphi method is only used to forecast, put forward the index system and determine the specific index.

2. In the research design section, the COVID-19 assessment scorecard consists of 19 policy statements [1], but the author noted 20 in the manuscript.

3. It is odd for the authors to use the Cambridge Dictionary's definition of scholarship to select a study population. The views of the health sector and healthcare providers on policies to control COVID-19 are even more important. The study was underrepresented.

4. In the results section, it is more intuitive to use the expert status table. For example, age can be expressed in three groups.

5. Please explain in the methods section why the P value is calculated. In general, the Delphi method does not require the calculation of P- values.

6. It is too tedious to present all policy statements in the results. Simplify them.

7. From the results of this study, moderate consensus was reached on only 5 statements, low consensus was reached on 13 statements, and no consensus was reached on 2 statements. In the Delphi method, the reliability of the results should be proved. However, the expert' authority coefficient (Cr) and the coefficient of variation (CV) are necessary. The reliability and representativeness of the manuscript are worrisome.

[1] Lazarus JV, Binagwaho A, El-Mohandes AAE, Fielding JE, Larson HJ, Plasència A, et al. Keeping governments accountable: the COVID-19 Assessment Scorecard (COVID- SCORE). Nat Med. 2020 Jul;26(7):1005-1008.

Reviewer #6: The authors present an important application of expert elicitation to guide policy decision making in the context of COVID-19. The methods for the expert elicitation are clearly reported, and the results are presented in a well-organized manner. The authors have responded satisfactorily to the comments from both rounds of reviewers. I have a couple of minor comments.

First, pg 15, please specify in text that no statements received a high consensus level.

Second, line 318, the statement “the government should involve domestic private manufacturers” is phrased differently from all other scorecard descriptions which state whether the experts think the government has done the task. Could the authors please clarify if this should in fact be past tense (has) or if future tense (should) is correct. If the latter, then please clarify why the interpretation for this statement is different.

Finally, I saw no mentions to the supplemental materials made in text. Please add references to these materials in the requisite sections. Additionally, the code book submitted in appendix 3 appears to be incomplete with descriptions absent and titles that do not correspond to the data elements in the data file. Have the authors considered attaching the complete survey as an additional supplement?

7. PLOS authors have the option to publish the peer review history of their article (what does this mean?). If published, this will include your full peer review and any attached files.

Reviewer #3: No

Reviewer #4: **Yes: **Redwan Bin Abdul Baten

Reviewer #5: No

Reviewer #6: No

---

## [Author Response · Author response to Decision Letter 2]

28 Jun 2022

Dear Editor,

Thank you for the reviewers’ comments. Please find below our point-by-point responses.

Best regards,

Yodi Mahendradhata

Reviewer #3: Overall Impressions:

All major issues have been catered for by the co-authors. There are however a number of minor issues, mainly typographical and grammatical, that the co-authors would need to address:

Issues:

Location Issue Line 19 ‘government’s’ was repeated twice

This has been corrected as follow:

“We aimed to investigate how medical and health academics view Indonesian government's handling of COVID-19 and which areas of health systems need to be prioritized to improve the government's response to COVID-19.”

Line 19 Remove ‘the’ in ‘…government’s handling of the COVID-19…’

This has been corrected as follow:

“We aimed to investigate how medical and health academics view Indonesian government's handling of COVID-19 and which areas of health systems need to be prioritized to improve the government's response to COVID-19.”

Line 25 Remove ‘have’ in the statement: ‘All participants who have completed…’

This has been corrected a follow:

“All participants who completed the first cycle were invited to participate in the second cycle.”

Line 27 ‘rank’ should be ‘ranked’

This has been corrected as follow:

“They had the opportunity to revise their answers based on the previous cycle's results and ranked a list of actions to improve government response.”

Replace ‘priority with ‘list’

This has been corrected as follow:

“They had the opportunity to revise their answers based on the previous cycle's results and ranked a list of actions to improve government response.”

Line 31 Replace ‘strengthen’ with ‘strengthening’

This has been corrected as follow:

“strengthening capacity to ensure consistent, credible and targeted communication while adopting a more inclusive and empathic communication style to address public concerns”

Line 33 Replace ‘ensure’ with ‘ensuring’

This has been corrected as follow:

“ensuring universal access to reliable COVID-19 test by expanding lab infrastructure, facilitating operational readiness, and scaling up implementation of proven alternative/complementary tests to RT-PCR;”

Line 35 Replace ‘boost’ with ‘boosting’

This has been corrected as follow:

“boosting contact tracing implementation capacity and facilitating contact tracing for all positive cases, involving key stakeholders in further development of the existing contact tracing system (i.e. PeduliLindungi) as well as its evaluation and quality assurance.”

Line 36 Replace ‘facilitate’ with ‘facilitating’

This has been corrected as follow:

“boosting contact tracing implementation capacity and facilitating contact tracing for all positive cases, involving key stakeholders in further development of the existing contact tracing system (i.e. PeduliLindungi) as well as its evaluation and quality assurance.”

Line 122 Replace ‘has’ with ‘had’

This has been corrected as follow:

“To be considered a participant in this study, each candidate had to fulfil the following criteria”

Line 132 Replace ‘university’ with ‘respective universities’

This has been corrected as follow:

“To ensure that respondents have the capacity and capability to participate in the survey, we then performed snowball sampling by asking senior lecturers to recommend a list of academics from their respective universities.”

Line 176 Replace ‘will be’ with ‘was’

This has been corrected as follow:

“For the ranking score, each priority action was be scored according to its sum of rank.”

Line 177 Replace ‘will be’ with ‘was considered’

This has been corrected as follow:

The lowest total score was considered the highest priority.

Line 188 Replace ‘can’ with ‘could’

This has been corrected as follow:

“Therefore, they could freely submit their ideas, unbiased by the identities or pressures of others.”

Line 190 Replace ‘are’ with ‘were’

This has been corrected as follow:

“All participants were coded in the analysis; thus, their opinions and comments were anonymous to the investigators.”

Line 225 Replace ‘has’ with ‘have’

This has been corrected as follow:

“Some of the results of this subsequent study have now been published elsewhere”

Line 289 Capitalize ‘low’

This has been corrected as follow:

“Low consensus level was achieved for the following statements:…”

Lines 290, 292, 298, 300, 301, 302, 304, 305 Replace ‘,’ with ‘;’

This has been corrected as follow:

“….the government has tried to maintain robust epidemiological databases at national and local levels (71.4% agree); the government should involve in country private manufacturers to produce critical equipment rapidly as part of corporate social responsibility (67.1% agree); a pandemic preparedness team that includes public health and medical experts is coordinating the national response (55.7% agree); infection prevention and care guidelines and protocols are comprehensive and up to date (61.4% agree); task sharing, task shifting and telehealth are being used to optimize the delivery of health care services (52.9% agree); public health measures have been implemented to protect people in institutions and other confined settings (55.0% agree); the government has target the entire population (68.5% disagree), there were enough qualified healthcare workers and medical equipment to meet national needs (67.1% disagree); the government has communicated clearly and consistently about COVID-19 (51.4% disagree); the government has enough funding and infrastructure to care for all COVID-19 patients, particularly in the long run (64.3% disagree); public health experts, government officials, and academic researchers agree on COVID-19 nomenclature and have clearly explain the reason for public health measures (54.2% disagree); access to regular health services was uninterrupted (64.3% disagree); and that appropriate measures have been taken to protect members of vulnerable groups, such as the elderly, the poor, migrants, and the homeless (61.4% disagree).”

Line 300 Replace ‘communicate’ with ‘communicated’

This has been corrected as follow:

“the government has communicated clearly and consistently about COVID-19 (51.4% disagree);”

Line 301 Replace ‘have’ with ‘has’

This has been corrected as follow:

“the government has enough funding and infrastructure to care for all COVID-19 patients, particularly in the long run (64.3% disagree)”

Line 303 Replace ‘has’ with ‘have’

This has been corrected as follow:

“public health experts, government officials, and academic researchers agree on COVID-19 nomenclature and have clearly explain the reason for public health measures (54.2% disagree);”

Line 310 Replace ‘eight’ with ‘eighth’

This has been corrected as follow:

“It ranked eighth on priority recommended actions.”

Line 337 Modify ‘at what extent does’ to ‘the extent to which’

This has been corrected as follow:

“Thus, the extent to which public health experts have been involved in the response team remains questionable.”

Line 359 Replace ‘needs’ with ‘needed’

This has been corrected as follow:

“Payment for telehealth services was also another issue that needed to be discussed by policymakers.”

Line 394 Replace ‘suggest’ with ‘suggested that’

This has been corrected as follow:

“Respondents suggested that campaigns should adapt local wisdom and use language that should be easy for people with low literacy to understand.”

Line 396 Modify ‘rural’ to read ‘rural areas’

This has been corrected as follow:

“The use of the internet and social media to share information has discriminated population living in rural areas without access to the internet or television.”

Line 410 Replace ‘issue’ with ‘issues’

This has been corrected as follow:

“There were also issues regarding terms used for infection control, disease transmission and treatment.”

Line 417 Modify ‘price skyrocketing’ to read ‘prices were skyrocketing’

This has been corrected as follow:

“Medical personal protective equipment was scarce in every part of the country, and prices were skyrocketing after months of COVID-19.” 

Line 446 Replace ‘philanthropic’ with ‘philanthropies’

This has been corrected as follow:

“Many non-governmental organizations and philanthropies raise funds to support those affected by the COVID-19 pandemic, including the poor, elderly, and frontline healthcare workers.”

Line 504 Replace ‘contradict’ with ‘contradicted

This has been corrected as follow:

“However, when the number of cases increased in several areas, mayors and governors also issued statements, which often contradicted the official data held by the presidential office and Minister of Health”

Line 531 Replace ‘dan’ with ‘and’

This has been corrected as follow:

“The Ministry of Communications and Informatics monitors the spread of COVID-19 related mis/dis-information daily, with records of 1,550 hoaxes related to Covid-19 and 177 hoaxes related to Covid-19 vaccines from January 2020 to April 2021”

Reviewer #4: Line 15: Please make sure to use one common tense (either present or past) throughout the abstract. There are multiple grammatical mistakes in this paragraph. Please correct them.

The abstract has been thoroughly checked with Grammarly Premium and corrected accordingly

Line: 19: Please omit the repeated words.

This has been corrected as follow:

Line 23: Please clarify how many rounds of data collection was performed.

The sentence has been revised as follow: 

“We invited medical and health academics from ten universities across Indonesia to take part in the two-round Delphi study”

Line 46: Please be consistent when characterizing the COVID-19 as a pandemic. Here the authors have depicted it as an epidemic, which may confuse the readers.

The sentence has now been revised as follow:

“Governments' failures to suppress the epidemic pandemic is disappointing and costly.”

Line 60: Please explain the term "zero case" - is it an academic term? If so, please provide a reference for it - otherwise please explain clearly for the readers.

The sentence has now been revised as follow:

“The initial claim of no case reported by Indonesia, before the first two confirmed cases, was questioned by many”

Line 62: Should it be specific quarantines, or specific policies/instructions/guidelines for quarantines? Please clarify.

The sentence has now been revised as follow:

“During this period, the government did not issue travel restrictions and specific policies for quarantines of travellers coming in/coming back to Indonesia, despite reports of increasing COVID-19 cases in neighbouring countries”

Line 64: Is there any data regarding Indonesian nationals versus tourists arriving in the country during that period of time? Please provide references as Indonesia is a known tourist country.

We could not find the requested data

Line 67: Please provide information on the first COVID-19 related death occurring in Indonesia - when did it occur and how did that impact the governments' decisions?

The sentence has now been revised as follow:

“After the initial and subsequent reports of infections as well as the first death reported from coronavirus on March 11, 2020, the Indonesian government realized the seriousness of the situation [8, 9].”

Line 75: When the authors mention "Indonesia's vulnerable health system" - what do they mean? Please provide some context as to why the healthcare system is vulnerable in Indonesia.

The sentence has now been revised as follow:

“The impact on Indonesia's vulnerable health system, due to insufficient health workforce and health care infrastructure, will be devastating if COVID-19 continues the trajectory observed in other countries.”

Line 76: Please clarify what the non-COVID-19 related impacts are.

This has now been revised as follow:

“Meanwhile, the economic, social, and non-COVID-19 related health system impact (e.g. disruptions of non-COVID-19 related health services) has already taken a significant toll on Indonesia.”

Line 78: Please be consistent with WHO announcements on COVID-19 being a pandemic.

The sentence has now been revised as follow:

“Coordinated and comprehensive actions to suppress the COVID-19 epidemic pandemic in Indonesia need to be enhanced.”

Line 82: Please clarify the difference between health professionals (line 80) and health academics (line 82). Please use one term consistently throughout the paper.

The sentences have been revised as follow:

“This study aims to consolidate advice from medical and health academics for the government to enhance the COVID-19 response in Indonesia. The study objectives are twofold. First, investigate how medical and health academics view the government of Indonesia's handling of the COVID-19 epidemic.”

Line 91: A better term that can be used here is "to achieve the study objectives".

The sentence has now been revised as follow:

“We conducted a modified Delphi study to address achieve the study objectives”

Line 96: Please correct for double spaces throughout the manuscript. Please use single spaces.

In accordance to PLOS One guideline, we retain the double-spaced manuscript text.

Line 102: If possible, please provide references of other studies that have used a similar methodology to study the COVID-19 pandemic.

We have not found other studies using the same method for the COVID-19 pandemic

Line 104: Please clarify what the "modified Delphi study" is. Is this different from the original method? If so, how and why is it different? Please clarify for the readers.

We’ve inserted the following additional sentence for clarity as requested:

“The modified Delphi method we used is similar to the standard Delphi method in terms of procedure (i.e., a series of rounds with selected experts) and intent (i.e., to arrive at consensus). The major modification consists of beginning the process with a set of carefully selected items.”

Line 114: Please provide some context as to what type of "adopted statements" were utilized in the study. Are these statements from the Indonesian government? If so, which government body issued them and in what context? Was there a specific time frame from which the statements were collected? What was the inclusion criteria for these statements? Please clarify.

The sentence has now been revised as follow:

“The investigators in this study reviewed the adopted COVID-SCORE statements, assessed content validity, construct validity and approved changes.” 

Line 126: Why is regional representation important in this context? Please clarify.

The sentence has been revised as follow:

“We aimed to have regional representativeness of academics in the panel as Indonesia is diverse across its regions.”

Line 127: When describing the sampling technique, please use more academic/formal terms such as purposive sampling etc.

The sentence has now been revised as follow:

“Thus, we started with purposively selecting sampling potential universities to represent four regions of Indonesia: Sumatera, Java, Kalimantan, and Eastern Indonesia.”

Line 132: Did the authors follow a snowball sampling technique in the second stage of sampling? Please clarify the different stages that were used during the sampling process.

The sentence has now been revised as follow:

“To ensure that respondents have the capacity and capability to participate in the survey, we then performed snowball sampling by asking senior lecturers to recommend a list of academics from their respective universities.”

Line 134: Was the data collection process completed online? Please clarify this at the beginning of the methods section. Otherwise, it is confusing to the reader.

The data collection was online and this has been mentioned in the data collection part of the method section as follow:

“The email contained a short description of the background and aims of the study, the process of two-cycle Delphi surveys, and a link to an online questionnaire on the Qualtrics Research Suite platform.”

Line 137: How was this number (about 30) determined? Please clarify.

We have added the following explanation and reference to the Study Population part of the method section”

“However, a Delphi panel usually consists of 15 to 30 participants from the same discipline, or five to 10 per category from different professional groupings [16].” 

Line 148: Please clarify why 2 languages were used in the survey instrument.

The sentence has now been revised as follow:

“All statements were written in Bahasa Indonesia and English to minimize variations in interpretations among participants.”

Line 159: Why was not the data de-identified? Why did the researcher have access to the identity of the survey participants? Please clarify.

All experts in the panel were kept anonymous to each other but not to the researcher throughout the process. The researchers need to be able to trace responses of the experts in order to be able to link responses across the rounds

Line 163: Please clarify - was the feedback shared with the participants?

As stated at the end of the data collection part of the method section: 

“The result of the second (final) cycle was analyzed, and feedback to all participants a week later.”

Line 167: Please correct the spelling for calculated.

This has now been corrected as follow:

“We calculated percentages of score for each statement.”

Line 170: Please clarify how this decision criteria was determined. Please provide relevant references if possible.

The following sentence and reference has been added:

“We used percent agreement, which is the most common definition for consensus in Delphi [17].”

The 75% threshold set for Moderate consensus level is derived from median threshold of previous delphi surveys as noted in the reference.

Ref: Diamond IR, Grant RC, Feldman BM, Pencharz PB, Ling SC, Moore AM, Wales PW. Defining consensus: a systematic review recommends methodologic criteria for reporting of Delphi studies. J Clin Epidemiol. 2014 Apr;67(4):401-9. doi: 10.1016/j.jclinepi.2013.12.002.

Line 180: Please clarify what is meant by 1,5 hours.

This has now been revised as follow:

“The webinar was delivered via Zoom, facilitated by the authors and completed in 90 minutes”

Line 181: In case a respondent disagreed with the results - how was that handled by the researchers? What implications might it have for the results of the study?

If a respondent disagreed then the results would have to be discussed with the panellists to decide the final level of agreement

Line 189: Earlier it was mentioned that the researchers/authors had access to participant identities. This could have affected the responses from the participants. How was this handled by the authors?

To minimize biases, we employed standard techniques that are recommended in delphi surveys, eg. Include reasons in controlled feedback; Conduct multiple rounds of surveys.

Ref: Hallowel MR. Techniques to Minimize Bias When Using the Delphi Method to Quantify Construction Safety and Health Risk. Construction Research Congress 2009. Published online: April 26, 2012. https://doi.org/10.1061/41020(339)151. 

Line 287: Was this a recommendation made by the respondents?

Yes, this recommendation was derived from respondents’ responses

Line 325: This can be better expressed as quality of product.

This has now been revised as follow:

“Thus, guidance and supporting policy are needed to attract investors to collaborate with the government and ensure public trust in products quality of products.”

Line 337: Please correct grammatical issues with this sentence.

This has now been revised as follow:

“Thus, the extent to which does public health experts have been involved in the response team remains questionable.”

Line 344: Please provide a reference for this guideline.

The following reference has now been added:

Ministry of Health. COVID 19 Infection prevention and control guideline. 5th Ed. Jakarta: Ministry of Health Republic of Indonesia; 2020.

Line 355: Please rewrite this sentence to make it clear for the readers.

This has now been revised as follow:

“However, respondents were concerned that a policy to guide the implementation of a wide range of telehealth services was unavailable”

Line 376: Please provide references for these resources.

The following reference has been added as requested:

Sukarsa, AR. Devi OS. Analysis of Indonesian Government Public Communication based on Image Restoration Theory. Jurnal Mantik. 2021;5(2):1192-1199

Line 410: Please correct grammatical issues with this sentence.

This has now been revised as follow:

“There were also issues regarding terms used for infection control, disease transmission and treatment.”

Line 446: Please correct grammatical issues with this sentence.

This has now been revised as follow:

Many non-governmental organizations and philanthropies raise funds to support those affected by the COVID-19 pandemic, including the poor, elderly, and frontline healthcare workers.”

Line 451: Are these 2 separate departments of the government? Please clarify.

Yes, these are separate ministries

Line 470: Overall, the discussion section should be made more concise, as several paragraphs can be summarized into smaller ones.

We have revised the discussion section to make it more concise

Line 471: The results mentioned here has already been mentioned in the results section. This paragraph is thus no longer needed.

The results have indeed been mentioned on the results section, but we believe it’s useful to revisit the key results briefly at the beginning of the discussion section to reorient the readers.

Line 478: The results mentioned here has already been mentioned in the results section. This paragraph is thus no longer needed.

The results have indeed been mentioned on the results section, but we believe it’s useful to revisit the key results briefly at the beginning of the discussion section to reorient the readers.

Line 513: Please summarize the discussion on communication in a more concise way.

The discussion on communication has now been revised to be more concise

Line 612: Please clarify the meaning of higher levels of evidence.

This refers to the notion of hierarchy of evidence, in such context this could encompass, e.g. Descriptive surveys, qualitative studies, systematic review, metasynthesis, etc

Reviewer #5: Thanks for the authors for introducing us how medical and health academics view the Indonesian government's handling of the COVID-19 and which areas of health systems need to be prioritized. However, there are several concerns:

1. Although the author explains the reasons for using the Delphi method in page 5 lines 93-95, this kind of research does not require the Delphi method (time-consuming and laborious). The same result would have been obtained if a questionnaire had been used. The commonly used Delphi method is only used to forecast, put forward the index system and determine the specific index.

The delphi technique is commonly used in developing consensual guidance, not necessarily used to forecast. A traditional survey would have been difficult to conduct during the pandemic. An online survey not using the delphi method would require much more respondents and would get low response due to less engagement

2. In the research design section, the COVID-19 assessment scorecard consists of 19 policy statements [1], but the author noted 20 in the manuscript.

The research design section as follow mention 20 statements: 

“The COVID-SCORE consists of twenty policy statements about improving public health communication and health literacy, facilitating robust surveillance and reporting, developing pandemic preparedness, strengthening the health system, ensuring the health and social equity, and ensuring comprehensive confinement and de-confinement strategies. The twenty statements from COVID-SCORE were then modified into three types of questionnaires: Likert scale survey, short comment, and ranking scale.”

3. It is odd for the authors to use the Cambridge Dictionary's definition of scholarship to select a study population. The views of the health sector and healthcare providers on policies to control COVID-19 are even more important. The study was underrepresented.

The Delphi method is primarily used to elicit opinions of experts, thus the scholarship definition is highly relevant

4. In the results section, it is more intuitive to use the expert status table. For example, age can be expressed in three groups.

We feel that the descriptive characterization of the experts is sufficient

5. Please explain in the methods section why the P value is calculated. In general, the Delphi method does not require the calculation of P- values.

This has been explained in the method section as follow:

“We calculated p-values to assess the significance of the change in the degree of consensus for each statement between the cycles”

6. It is too tedious to present all policy statements in the results. Simplify them.

We believe it is important to present all policy statements in the results

7. From the results of this study, moderate consensus was reached on only 5 statements, low consensus was reached on 13 statements, and no consensus was reached on 2 statements. In the Delphi method, the reliability of the results should be proved. However, the expert' authority coefficient (Cr) and the coefficient of variation (CV) are necessary. The reliability and representativeness of the manuscript are worrisome.

We follow the Guidance on Conducting and Reporting DElphi Studies (CREDES). This guidance does not call for calculating and reporting of expert' authority coefficient (Cr) and the coefficient of variation (CV) 

Reviewer #6: The authors present an important application of expert elicitation to guide policy decision making in the context of COVID-19. The methods for the expert elicitation are clearly reported, and the results are presented in a well-organized manner. The authors have responded satisfactorily to the comments from both rounds of reviewers. I have a couple of minor comments.

First, pg 15, please specify in text that no statements received a high consensus level.

This has now been incorporated in the second paragraph of the results section as follow:

“Out of 20 survey statements, we achieved a moderate consensus level for five statements, 13 low consensus levels, and no consensus for two statements. No statements received a high consensus level.”

Second, line 318, the statement “the government should involve domestic private manufacturers” is phrased differently from all other scorecard descriptions which state whether the experts think the government has done the task. Could the authors please clarify if this should in fact be past tense (has) or if future tense (should) is correct. If the latter, then please clarify why the interpretation for this statement is different.

This has now been revised as:

“Respondents generally agreed that the government can require private manufacturers to produce critical equipment rapidly. The government has involved domestic private manufacturers to produce critical equipment rapidly”

Finally, I saw no mentions to the supplemental materials made in text. Please add references to these materials in the requisite sections. Additionally, the code book submitted in appendix 3 appears to be incomplete with descriptions absent and titles that do not correspond to the data elements in the data file. Have the authors considered attaching the complete survey as an additional supplement?

We’ve added the following sentence to the data collection part of the method section:

“The survey questionnaire is available from the authors on request”

We’ve also added the following sentence to the analysis part of the method section

“The dataset and code book has been made available as supplementary files to this manuscript.”

---

## [Decision Letter · Decision Letter 3]

12 Sep 2022

Strengthening government’s response to COVID-19 in Indonesia: a modified Delphi study of medical and health academics

PONE-D-20-34947R3

Dear Dr. Mahendradhata, 

We’re pleased to inform you that your manuscript has been judged scientifically suitable for publication and will be formally accepted for publication once it meets all outstanding technical requirements.

Within one week, you’ll receive an e-mail detailing the required amendments. You also need to correct the grammatical errors in your manuscript. When these have been addressed, you’ll receive a formal acceptance letter and your manuscript will be scheduled for publication.

Kind regards,

Rubeena Zakar, Ph.D

Section Editor

PLOS ONE

Additional Editor Comments (optional):

Reviewers' comments:

Reviewer's Responses to Questions

**Comments to the Author**

1. If the authors have adequately addressed your comments raised in a previous round of review and you feel that this manuscript is now acceptable for publication, you may indicate that here to bypass the “Comments to the Author” section, enter your conflict of interest statement in the “Confidential to Editor” section, and submit your "Accept" recommendation.

Reviewer #3: All comments have been addressed

Reviewer #4: All comments have been addressed

Reviewer #5: (No Response)

Reviewer #6: All comments have been addressed

2. Is the manuscript technically sound, and do the data support the conclusions?

Reviewer #3: Yes

Reviewer #4: Yes

Reviewer #5: Partly

Reviewer #6: Yes

3. Has the statistical analysis been performed appropriately and rigorously? 

Reviewer #3: Yes

Reviewer #4: Yes

Reviewer #5: No

Reviewer #6: Yes

4. Have the authors made all data underlying the findings in their manuscript fully available?

Reviewer #3: Yes

Reviewer #4: Yes

Reviewer #5: No

Reviewer #6: Yes

5. Is the manuscript presented in an intelligible fashion and written in standard English?

Reviewer #3: Yes

Reviewer #4: Yes

Reviewer #5: No

Reviewer #6: Yes

6. Review Comments to the Author

Reviewer #3: Overall Impressions:

I think this is a highly-improved version of the earlier drafts. I would like to commend the co-authors for being patient to incorporate the suggested modifications. I still have very minor suggestions – mainly grammar.

Location Issue

Line 17 Replace ‘continued’ with ‘continue’

Line 73 Replace ‘By’ with ‘As at’

Line 81 Replace ‘need’ with ‘needs’

I am for the publication of this manuscript. Thanks.

Reviewer #4: Thanks to the authors for addressing the comments/suggestions from all reviewers. The revisions have certainly improved the quality of the manuscript.

Reviewer #5: The author does not understand the relevant knowledge and writing method of Delphi method. Most of the questions I raised were evaded. This is not the status of a scientific worker. Therefore, I do not think this manuscript qualifies for publication in Plos One.

From the results of this study, moderate consensus was reached on only 5 statements, low consensus was reached on 13 statements, and no consensus was reached on 2 statements. The results are worrying. In addition, it is odd for the authors to use the Cambridge Dictionary's definition of scholarship to select a study population. The views of the health sector and healthcare providers on policies to control COVID-19 are even more important. The study was underrepresented.

So I have to reject it.

Reviewer #6: The authors have addressed all comments raised by the reviewers. I have a few minor grammatical suggestions.

Line 18: the word "the" was erronously deleted

Line 34: should read "COVID-19 testing"

Line 44: The authors may want to consider updating the number of COVID cases and deaths

Line 82: should read "the government needs" and the authors may wish to provide additional context as to why this is needed

Line 85 should read "First, to investigate"

Line 86-87: should read "Second, to identify"

Line 123: should read "academics who were not working in the institutions"

line 270: should read "tracing"

Line 272: shoudl read "the government had tried to address the health and"

Line 279: should read "there were no sanctions"

Line 289: should read "when people got tested"

Line 322: should read "the government had tried"

Line 345: should read "it ranked tenth"

Line 480: should read "Only one-third"

Line 505: should read "comunication was considered critical"

Line 509: the '' before Wuhan should be delete

Line 515: pandemic should not be capitalized

Line 120: delete "and periodically"

Line 537: should read "thus needs to improve"

Pg 30: the authors may wish to update the values from August 2021

7. PLOS authors have the option to publish the peer review history of their article (what does this mean?). If published, this will include your full peer review and any attached files.

Reviewer #3: No

Reviewer #4: No

Reviewer #5: No

Reviewer #6: No

---

## [Editor Report · Acceptance letter]

21 Sep 2022

PONE-D-20-34947R3 

Strengthening government’s response to COVID-19 in Indonesia: a modified Delphi study of medical and health academics 

Dear Dr. Mahendradhata:

I'm pleased to inform you that your manuscript has been deemed suitable for publication in PLOS ONE. Congratulations! Your manuscript is now with our production department. 

Kind regards, 

on behalf of

Dr. Rubeena Zakar 

Section Editor

PLOS ONE